

# A benchmark for multimodal fact verification with explainability through 5W Question-Answering

**Megha Chakraborty[1]  Khushbu Pahwa[2]  Anku Rani[1]  Shreyas Chatterjee[1]  Dwip Dalal[1]**
**Harshit Dave[1]  Ritvik G[1]  Preethi Gurumurthy[1]  Adarsh Mahor[1]  Samahriti Mukherjee[1]**
**Aditya Pakala[1]  Ishan Paul[1]  Janvita Reddy[1]  Arghya Sarkar[1]  Kinjal Sensharma[1]**
**Aman Chadha[3,4]†  Amit P. Sheth[1]  Amitava Das[1]**

[1] University of South Carolina, USA  [2] UCLA, USA  [3] Amazon AI, USA  [4] Stanford University, USA

meghac@email.sc.edu   amitava@mailbox.sc.edu

## Abstract

Combating disinformation is one of the burning societal crises - about 67% of the American population believes that disinformation produces a lot of uncertainty, and 10% of them knowingly propagate disinformation. Disinformation can manipulate democracy, public opinion, disrupt markets, and cause panic or even fatalities. Thus, swift detection and possible prevention of disinformation are vital, especially with the daily flood of 3.2 billion images and 720,000 hours of videos on social media platforms, necessitating efficient fact verification. Despite progress in automatic text-based fact verification (e.g., FEVER, LIAR), the research community lacks substantial effort in multimodal fact verification. To address this gap, we introduce FACTIFY 3M, a dataset of 3 million samples that pushes the boundaries of the domain of fact verification via a multimodal fake news dataset, in addition to offering explainability through the concept of 5W question-answering. Salient features of the dataset are: *(i) textual claims, (ii) GPT3.5-generated paraphrased claims, (iii) associated images, (iv) stable diffusion-generated additional images (i.e., visual paraphrases), (v) pixel-level image heatmap to foster image-text explainability of the claim, (vi) 5W QA pairs, and (vii) adversarial fake news stories.*

---

†Work does not relate to the position at Amazon.

## 1   FACTIFY 3M: An Illustration

We introduce FACTIFY 3M (*3 million*), the largest dataset and benchmark for multimodal fact verification.

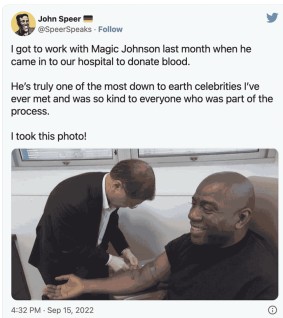

Figure 1: A tweet referring to the sports personality and known AIDS victim Magic Johnson with a photo that was taken a decade before COVID and, moreover, is not real evidence of blood donation.

Consider the example in Fig. 1. A widely distributed image of the sports legend Magic Johnson with an IV line in his arm was accompanied by the claim that he was donating blood, implicitly during the COVID-19 pandemic. If true, this is troubling because Magic Johnson is a well-known victim of AIDS and is prohibited from donating blood. The picture predated the COVID-19 epidemic by a decade and is related to his treatment for AIDS.

**Textual claim:** The text associated with the claim in Fig. 1 purports that the tweet's author took this photo and assisted Magic Johnson in donating

| | Entailment classes | Textual Support | Visual/Image Support | No. of claims | No. of paraphrased claims | No. of images | No. of stable diffusion generated images | 5WQA pairs | No. of evidence documents | Adversarial OPT-generated news story |
|---|---|---|---|---|---|---|---|---|---|---|
| | **PromptFake3M at a glance** | | | | | | | | | |
| Support | Support_Multimodal | Texts are supporting each other ∼similar news | Images are supporting each other | 232,000 | 882,000 | 232,000 | 927,000 | 858,400 | 232,000 | |
| Support | Support_Text | Texts are supporting each other ∼similar news | Images are neither supporting nor refuting | 174,000 | 609,000 | 169,000 | 661,000 | 852,600 | 174,000 | |
| Neutral | Insufficient_Multimodal | Texts are neither supported nor refuted ∼may have common words | Images are supporting each other | 99,000 | 366,000 | 99,000 | 347,000 | 375,000 | 99,000 | |
| Neutral | Insufficient_Text | Texts are neither supported nor refuted ∼may have common words | Images are neither supporting nor refuting | 126,000 | 525,000 | 123,000 | 466,000 | 441,000 | 126,000 | |
| Fake | Refute | Fake claim | Fake image support | 316,000 | 1,193,000 | 309,000 | 916,400 | 1,327,000 | 316,000 | 135,000 |
| | Total | | | 947,000 | 3,575,000 | 932,000 | 3,317,400 | 3,954,000 | 947,000 | 135,000 |

Table 1: A top-level view of FACTIFY 3M: (i) classes and their respective textual/visual support specifics, (ii) number of claims, paraphrased claims, associated images, generated images, 5W pairs, evidence documents, and adversarial stories.

Figure 2: An illustration of the proposed 5W QA-based explainable fact verification system. This example illustrates the false claim shown in Fig. 1. A typical semantic role labeling (SRL) system processes a sentence and identifies verb-specific semantic roles. Therefore, for the specified example, we have 3 sentences: sentence 1 has two main verbs *work* and *come*, sentence 2 has one verb *meet*, and sentence 3 has one verb *take*. For each verb, a 5W QA pair will be automatically generated (4 × 5 = 20 sets of QA pairs in total for this example). Furthermore, all those 20 5W aspects will be fact-checked. If some aspects end up having *neutral* entailment verdict, possible relevant documents with associated URLs will be listed for the end user to read further and assess. In addition, a reverse image search result will be shown to aid human fact-checkers further.

Figure 3: Claims paraphrased using GPT3.5 (Brown et al., 2020) to foster their textual diversity.

Figure 4: An example of OPT (Zhang et al., 2022) generated fake news that confirms the Magic Johnson blood donation incident.

blood. The further implicit declaration is that he is a medical worker and possibly works for a hospital that Magic Johnson visited for blood donation.

**GPT3.5-paraphrased claims:** To emulate the textual diversity in how news publishing houses report day-to-day happenings, it is important to pro-duce data representative of such variety. Variety in text is embodied through a difference in narrative styles, choice of words, ways of presenting factual information, etc. For e.g., Fig. 5a shows a claim and document that address the same topic but differ in their textual prose. To mimic this aspect, we

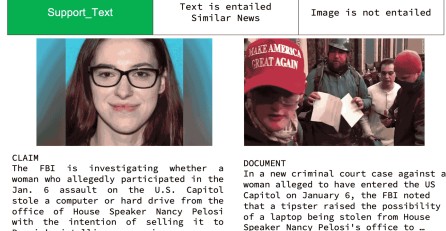

(a) Another example of covering the same news event by two news media houses. Here the same alleged lady is visible in both images, but altogether two images are different, and the text is paraphrased differently.

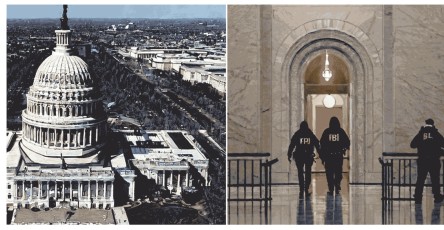

(b) Stable Diffusion output for the above claim.

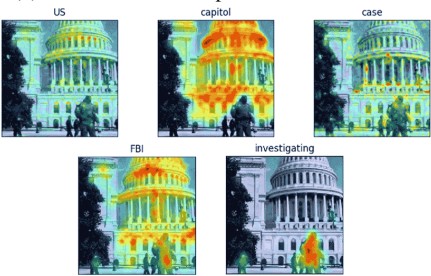

(c) DAAM (Tang et al., 2022) explanation for the above claim.

Figure 5: An example from PromptFake 3M dataset. Multimodal fact verification is a challenging endeavor considering the intricacies of real-life data, where the entailment of images requires understanding the nuances of day-to-day life situations. As such, multimodal entailment is an under-research paradigm with limited progress; current SoTA systems lack the finesse needed to handle the complexity portrayed in these previous examples adeptly.

adopted GPT3.5 as a paraphraser and generated claims Fig. 3.

**Associated images:** The image included as part of the claim (refer Fig. 1 for the image embedded in the tweet and Fig. 5a for images included as part of the claim) improves its trustworthiness perception since humans tend to believe visual input much more than mere text prose. Moreover, the text and image components together provide a holistic claim assertion, similar to how news articles convey world updates.

**Stable Diffusion-generated additional images a.k.a. visual paraphrases:** Referring to Fig. 5a, the diversity of images associated with the claim and document are apparent. Specifically, the image associated with the claim is that of a frontal face, while that associated with the document, while still the same person, is one of a not-so-visible face but includes multiple other entities. Such diversity is commonly seen in the wild when different news media houses cover the same news. As such, we try to emulate this aspect in our work by harnessing the power of the latest in text-to-image generation. We generate additional images using Stable Diffusion (Rombach et al., 2021). Fig. 5b shows the diversity of generated images in terms of the camera angle, subject, entities, etc., which in turn offers enhanced visual diversity for a multimodal claim.

**Pixel-level image heatmap:** To clearly delineate and hence explain which aspect of the image is being referred to in the various components of the text caption, we generate pixel-level attribution heatmaps to foster explainability. For e.g., referring to Fig. 5c, the heatmap highlights gothic-architecture buildings for the word *capitol* which the building is known for, and highlights the human figure for the term *investigating*. A dataset of this kind would be very helpful in designing explainable multimodal fact verification and possibly visual question-answer-based fact verification.

**5WQA:** The process of fact verification is inherently intricate, with several questions representing the components within the underlying claim that need answers to reach a verdict on the veracity of the claim. Referring to the example in Fig. 1, such questions may include: *(a) who donated blood? (b) when did he donate blood? (c) can Magic Johnson donate blood? (d) what can go wrong if this claim is false?* Manual fact-checking can be labor-intensive, consuming several hours or days (Hassan et al., 2015; Adair et al., 2017).

Contemporary automatic fact-checking systems focus on estimating truthfulness using numerical scores, which are not human-interpretable. Others extract explicit mentions of the candidate's facts

in the text as evidence for the candidate's facts, which can be hard to spot directly. Only two recent works (Yang et al., 2022; Kwiatkowski et al., 2019) propose question answering as a proxy to fact verification explanation, breaking down automated fact-checking into several steps and providing a more detailed analysis of the decision-making processes. Question-answering-based fact explainability is indeed a very promising direction. However, open-ended QA for a fact can be hard to summarize. Therefore, we refine the QA-based explanation using the 5W framework (*who, what, when, where, and why*). Journalists follow an established practice for fact-checking, verifying the so-called 5Ws (Mott, 1942), (Stofer et al., 2009), (Silverman, 2020), (Su et al., 2019), (Smarts, 2017), (Wiki_Article, 2023). This directs verification search and, moreover, identifies missing content in the claim that bears on its validity. One consequence of journalistic practice is that claim rejection is not a matter of degree (as conveyed by popular representations such as a number of Pinocchios or crows, or true, false, half true, half false, pants on fire), but the rather specific, substantive explanation that recipients can themselves evaluate (Dobbs, 2012). Please refer to Fig. 2 to look at the possible 5W QA questionnaire for the claim in Fig. 1.

**Adversarial fake news:** Fact verification systems are only as good as the evidence they can reference while verifying a claim's authenticity. Over the past decade, with social media having mushroomed into the masses' numero-uno choice of obtaining world news, fake news articles can be one of the biggest bias-inducers to a person's outlook towards the world. To this end, using the SoTA language model, we generate adversarial news stories to offer a new benchmark that future researchers can utilize to certify the performance of their fact verification systems against adversarial news.

Programmatic detection of AI-generated writing (where an AI is the sole author behind the article) and its more challenging cousin – AI-assisted writing (where the authorship of the article is split between an AI and a human-in-the-loop) – has been an area of recent focus. While detecting machine-generated text from server-side models (for instance, GPT-3 (Brown et al., 2020), which is primarily utilized through an API, uses techniques like watermarking (Wiggers, 2022b)) is still a topic of investigation, being able to do so for the plethora of open-source LLMs available online is a herculean task. Our adversarial dataset will offer a testbed so that such detection advances can be measured against with the ultimate goal of curbing the proliferation of AI-generated fake news.

## 2 Related Works: Data Sources and Compilation

Automatic fact verification has received significant attention in recent times. Several datasets are available for text-based fact verification, e.g., FEVER (Thorne et al., 2018), Snopes (Vo and Lee, 2020), PolitiFact (Vo and Lee, 2020), FavIQ (Kwiatkowski et al., 2019), HoVer (Jiang et al., 2020), X-Fact (Gupta and Srikumar, 2021), CREAK (Onoe et al., 2021), FEVEROUS (Aly et al., 2021), etc.

Multimodal fact verification has recently started gaining momentum. DEFACTIFY workshop series at AAAI 2022 (Mishra, 2022) and 2023 (Suryavardhan, 2023) has released FACTIFY 1.0 (Mishra et al., 2022) and 2.0 (Mishra et al., 2023) with 50K annotated data each year, which we have embedded as well as part of FACTIFY 3M. Fact verification datasets are mainly classified into three major categories: *(i) support, (ii) neutral, and (iii) refute*. While it is relatively easier to collect data for *support* and *neutral* categories, collecting large-scale *refute* category fake news claims is relatively challenging. FEVER (Thorne et al., 2018) proposed an alternative via manual imaginary claim generation, but is complex, lacks scalability, and may generate something unrealistic. Therefore, we decided to merge available datasets, at least for the *refute* category. It is wise to have all these datasets into one, and further, we have generated associated images using stable diffusion. While

selecting datasets, we only chose datasets with evidence claim documents as we are interested in 5W QA-based explanation. Other datasets only with fake claims were discarded for this reason. Furthermore, we use OPT (Zhang et al., 2022) to generate adversarial fake news documents of text based on the refute claims as prompts.

We have adopted an automatic method to compile a list of claims for *support* and *neutral* categories. It is often seen that the same event gets reported by two different media houses separately on the same day - therefore, one can be treated as support for the other. With this strategy in mind, we have collected and annotated large-scale data automatically (cf. Appendix A for details). Fig. 6 visualizes how much data from each dataset are compiled.

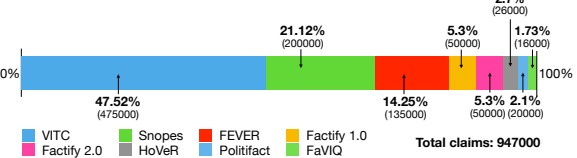

Figure 6: Distribution of our dataset delineating its constituent components.

Table 1 offers a statistical description of the five entailment classes, their definitions in terms of textual/visual support, an approximate count of the claims, paraphrased claims, images, 5W QA pairs, evidence documents, and adversarial stories for the Refute class. Furthermore, to solidify the idea behind the above categories, Fig. 5 offers a walkthrough of an example from the proposed dataset.

There are only a handful of other previous works, (Yao et al., 2022), (Abdelnabi et al., 2022), (Roy and Ekbal, 2021), (Nielsen and McConville, 2022), (Jin et al., 2017), (Luo et al., 2021), that have discussed multimodal fact verification. None of them generated large resources like ours and did not discuss QA-based explanation, heatmap-based image explainability, and adversarial assertion.

## 3 Paraphrasing Textual Claims

A claim may have multiple diverse manifestations depending on the style and manner in which it was reported. Specifically, the textual component

(i.e., prose) may have variations as highlighted in Fig. 5a. We seek to echo such heterogeneity to ensure the real-world applicability of our benchmark (cf. examples in Fig. 3 and more examples in the Appendix O). Manual generation of possible paraphrases is undoubtedly ideal but is time-consuming and labor-intensive. On the other hand, automatic paraphrasing has received significant attention in recent times (Sancheti et al., 2022; Xue et al., 2022; Bandel et al., 2022; Garg et al., 2021; Goyal and Durrett, 2020). Our criteria for selecting the most appropriate paraphrasing model was the linguistic correctness of the paraphrased output and the number of paraphrase variations. To achieve this, we propose the following process - let's say we have a claim $c$, we generate a set of paraphrases of $c$. Textual paraphrase detection is a well-studied paradigm, there are much state-of-the-art (SoTA) systems (Wang et al., 2021, 2019; Tay et al., 2021). We pick the best model available mostly trained based on the resources available from SNLI (Bowman et al., 2015). Next, we use the entailment model (Wang et al., 2019) to choose the right paraphrase candidate from the generated set, by doing a pairwise entailment check and choosing only examples which exhibit entailment with $c$. We empirically validated the performance of (a) Pegasus (Zhang et al., 2020), (b) T5 (Flan-t5-xxl variant) (Chung et al., 2022), and (c) GPT-3.5 (gpt-3.5-turbo-0301 variant) (Brown et al., 2020) models for our use-case and found that GPT-3.5 outperformed the rest (Appendix C for details on evaluation along three dimensions: (i) coverage, (ii) correctness, and (iii) diversity).

## 4 Visual Paraphrase: Stable Diffusion-based Image Synthesis

While textual diversity in claims seen in the wild is commonplace, typically the visual components – particularly, images – also show diversity. The concept of AI-based text-to-image generators has been around for the past several years, but their outputs were rudimentary up until recently. In the

past year, text prompt-based image generation has emerged in the form of DALL-E (Ramesh et al., 2021), ImageGen (Saharia et al., 2022), and Stable Diffusion (Rombach et al., 2021). While these new-age systems are significantly more powerful, they are a double-edged sword. They have shown tremendous potential in practical applications but also come with their fair share of unintended use cases. One major caution is the inadvertent misuse of such powerful systems. To further this point, we have utilized Stable Diffusion 2.0 (Rombach et al., 2021) to generate a large amount of fake news data.

Stable Diffusion (Rombach et al., 2021) is a powerful, open-source text-to-image generation model. The model is not only extremely capable of generating high-quality, accurate images to a given prompt, but this process is also far less computationally expensive than other text-conditional image synthesis approaches such as (Ding et al., 2021; Nichol et al., 2021; Zhou et al., 2021; Gafni et al., 2022). Stable diffusion works on stabilizing the latent diffusion process which has an aspect of randomization, as a result, it generates a different result each time. Moreover, quality control is a challenge. We have generated 5 images for a given claim and then further ranked them, discussed in the next section (cf. Appendix N for examples).

### 4.1 Re-ranking of Generated Images

In order to quantitatively assess and rank the images generated by the stable diffusion model, we leverage the CLIP model (Radford et al., 2021a) to obtain the best image conditioned on the prompt. We use CLIP-Score based re-ranking to select the best image corresponding to the prompt. The CLIP-Score denotes the proximity between the final image encodings and the input prompt encoding.

### 4.2 Pixel-level Image Heatmap

(Tang et al., 2022) perform a text–image attribution analysis on Stable Diffusion. To produce pixel-level attribution maps, authors propose Diffusion Attentive Attribution Maps (DAAM), a novel interpretability method based on upscaling and aggregating cross-attention activations in the latent denoising subnetwork. We adapt the official code available on Github (Castorini, 2022) to obtain the attribution maps in the generated images for each word in the cleaned prompt (pre-processed prompt after removal of stop-words, links, etc.). See Fig. 5c and Appendix N for examples.

### 4.3 Quality Assessment of Synthetically Generated Images

While SD has received great acclaim owing to its stellar performance for a variety of use cases, to our knowledge, to our knowledge, we are the first to adopt it for fake news generation. As such, to assess the quality of generated images in the context of the fake news generation task, we utilize two evaluation metrics.

We use Fréchet inception distance (FID) (Heusel et al., 2017) which captures both fidelity and diversity and has been the de-facto standard metric for SoTA generative models (Karras et al., 2018, 2019; Ho et al., 2020; Brock et al., 2018). The process we adopted to compute the FID score to quantitatively assess the quality of SD-generated images is detailed in E.1. For a chosen set of 500 claims (100 per category), we obtained an FID score. We obtained an FID score of 8.67 (lower is better) between the set of real images and SD-generated images for the same textual claims.

As our secondary metric, we utilized Mean Opinion Score (MOS) at the claim category level which is a numerical measure of the human-judged perceived quality of artificially generated media (cf. Appendix E.2 for process details). Results of the conducted MOS tests are summarized in Fig. 17.

## 5 Automatic 5W QA Pair Generation

A false claim is very likely to have some truth in it, some correct information. In fact, most fake news articles are challenging to detect precisely because they are mostly based on correct information, deviating from the facts only in a few aspects. That is, the misinformation in the claim comes from a very specific inaccurate statement. So, given our textual claim and image claim, we generate 5W

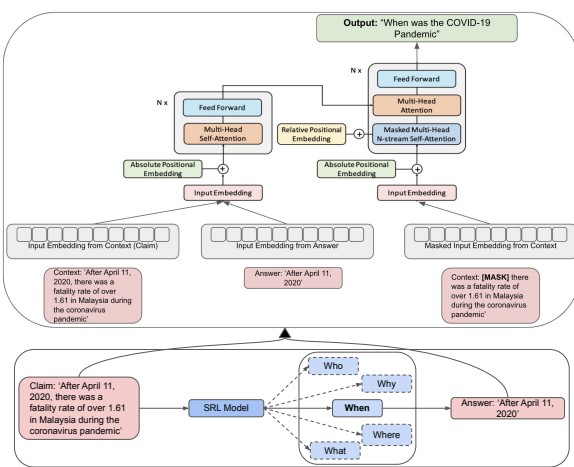

Figure 7: 5W QA Generation Pipeline using ProphetNet.

question-answer pairs by doing semantic role labeling on the given claim. The task is now based on the generated QA pairs, a fact-checking system can extract evidence sentences from existing authentic resources to verify or refute the claim based on each question- *Who, What, When, Where, and Why*. Example in the figure 19 and in Appendix O.

We leverage the method of using 5W SRL to generate QA pairs (Rani et al., 2023) and verify each aspect separately to detect '*exactly where the lie lies*'. This, in turn, provides an explanation of why a particular claim is refutable since we can identify exactly which part of the claim is false.

### 5.1 5W Semantic Role Labelling

Identification of the functional semantic roles played by various words or phrases in a given sentence is known as semantic role labeling (SRL). SRL is a well-explored area within the NLP community. There are quite a few off-the-shelf tools available: (i) Stanford SRL (Manning et al., 2014), (ii) AllenNLP (AllenNLP, 2020), etc. A typical SRL system first identifies verbs in a given sentence and then marks all the related words/phrases haven relational projection with the verb and assigns appropriate roles. Thematic roles are generally marked by standard roles defined by the Proposition Bank (generally referred to as Prop-Bank) (Palmer et al., 2005), such as: *Arg0, Arg1, Arg2*, and so on. We propose a mapping mech-

anism to map these PropBank arguments to 5W semantic roles. The conversion table 4 and necessary discussion can be found in Appendix F.

### 5.2 Automatic 5W QA Pair Generation

We present a system for generating 5W aspect-based questions generation using a language model (LM) that is fed claims as input and uses the SRL outputs as replies to produce 5W questions with respect to the 5W outputs. We experimented with a variety of LMs: BART (Lewis et al., 2019) and ProphetNet (Qi et al., 2020), eventually settling on ProphetNet (see Fig. 7) based on empirically figuring out the best fit for our use-case (cf. Appendix G for details).

We then create answers using the evidence from the questions generated using ProphetNet by running them through T5 (Yamada et al., 2020) – a SoTA QA model, using the 5W-based generated questions. See Section 7.2 for details.

## 6 Injecting Adversarial Assertions

The rise of generative AI techniques with capabilities that mimic a human's creative thought process has led to the availability of extraordinary skills at the masses' fingertips. This has led to the proliferation of generated content, which is virtually indistinguishable as real or fake to the human eye, even for experts in some cases. This poses unparalleled challenges to machines in assessing the veracity of such content.

As one of the novelties of our work, we address this by introducing synthetically generated adversarial fake news documents for all the *refute* claims using OPT (Zhang et al., 2022), a large language model. In doing so, we attempt to confuse the fact verification system by injecting fake examples acting as an adversarial attack. To draw a parallel in a real-world scenario, this could mean the proliferation of such fake news articles online via social media, blog posts, etc. which would eventually lead to a fact-verification system being unable to make a concrete decision on the trustworthiness of the news. Such a scenario would lend itself as a natural manifestation of an adversarial attack by virtue (rather, the "vice") of the fake news articles confusing the

fact verification system. We analyze the impact on the performance of our fact verification system in table 2. Our goal in offering these adversarial articles as one of our contributions is to provide future researchers a benchmark using which they can measure (and hence, improve) the performance of their fact verification system.

## 6.1 Accuracy of Text Generation

We assess the quality of text generation using perplexity as an evaluation metrics. Perplexity is a measure of the likelihood of the generated sentence on a language model. We use a pre-trained GPT-2 model to evaluate text perplexity. A lower value is preferred. We have used the GPTZero detector to evaluate our perplexity score (Tian, 2023). Checking for paraphrased text generated over a 50 claims (25 original and 25 adversarial), we report an average perplexity score of 129.06 for original claims and 175.80 for adversarial claims ( Appendix J).

## 7 Experiments: Baselines & Performance

In this section, we present baselines for: *(i) multimodal entailment, (ii) 5W QA-based validation, and (iii) results of our models after adversarial injections of generated fake news stories.*

### 7.1 Multimodal entailment: *support or refute?*

In this paper, we model the task of detecting multimodal fake news as multimodal entailment. We assume that each data point contains a reliable source of information, called *document*, and its associated image and another source whose validity must be assessed, called the *claim* which also contains a respective image. The goal is to identify if the claim entails the document. Since we are interested in a multimodal scenario with both image and text, entailment has two verticals, namely textual entailment, and visual entailment, and their respective combinations. This data format is a stepping stone for the fact-checking problem where we have one reliable source of news and want to identify the fake/real claims given a large set of multimodal claims. Therefore the task essentially is: given a textual claim, claim image, text document, and document

image, the system has to classify the data sample into one of the five categories: Support_Text, Support_Multimodal, Insufficient_Text, Insufficient_Multimodal, and Refute. Using the Google Cloud Vision API (Google, 2022), we also perform OCR to obtain the text embedded in images and utilize that as additional input.

**Text-only model:** Fig. 16 shows our text-only model, which adopts a siamese architecture focussing only on the textual aspect of the data and ignores the visual information. To this end, we generate sentence embeddings of the claim and document attributes using a pretrained MPNet Sentence BERT model (Reimers and Gurevych, 2019a) (specifically the all-mpnet-base-v2 variant). Next, we measure the cosine similarity using the generated embeddings. The score, thus generated, is used as the only feature for the dataset, and classification is evaluated based on their F1 scores.

**Multimodal model:** Information shared online is very often of multimodal nature. Images can change the context of a textual claim (and vice versa) and lead to misinformation. As such, to holistically glean information from the available data, it is important that we consider both the visual and textual context when classifying the claims. Our multimodal architecture (Fig. 8), adopts a siamese architecture and utilizes both modalities. As we utilize an entailment-based approach, features from both the claim and document image-text pairs must be extracted. To this end, we utilize the pretrained MPNet Sentence BERT model (Reimers and Gurevych, 2019a) (specifically the all-mpnet-base-v2 variant) as our text embedding extractor and a pretrained Vision Transformer (ViT) model (Dosovitskiy et al., 2020) (specifically the vit-base-patch16-224-in21k variant) as our vision embedding extractor. The cosine similarity score is computed between both the claim and document image features. Furthermore, we also compute the cosine similarity for the text embeddings, similar to the text-only model.

Table 2 shows the F1 score for the unimodal (i.e., text-only) and multimodal approaches (cf.

| | Support_Text | | Support_Multimodal | | Insufficient_Text | | Insufficient_Multimodal | | Refute | | **Average** | |
|---|---|---|---|---|---|---|---|---|---|---|---|---|
| | Text-only | Multimodal | Text-only | Multimodal | Text-only | Multimodal | Text-only | Multimodal | Text-only | Multimodal | Text-only | Multimodal |
| Pre-adversarial attack (F1) | 0.33 | 0.61 | 0.15 | 0.60 | 0.22 | 0.58 | 0.22 | 0.57 | 0.31 | 0.65 | 0.25 | 0.60 |
| Post-adversarial attack (F1) | 0.15 (55% ↓) | 0.46 (25% ↓) | 0.06 (60% ↓) | 0.43 (28% ↓) | 0.21 (4% ↓) | 0.56 (3% ↓) | 0.11 (54% ↓) | 0.55 (43% ↓) | 0.11 (64% ↓) | 0.37 (43% ↓) | 0.13 (48% ↓) | 0.47 (21% ↓) |

Table 2: Results of the text-only and multimodal baselines pre- and post-adversarial attack.

pre-adversarial attack row in table 2) trained using their respective feature sets. The multimodal model shows a distinct improvement in performance compared to the text-only model, indicating the value-addition of the visual modality.

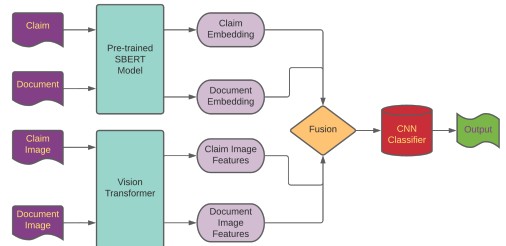

Figure 8: Multimodal baseline model which takes as input: (i) claim text, (ii) claim image, (iii) document text, and (iv) document image.

## 7.2 5W QA-based Validation

We generate 5W Question-Answer pairs for the claims, thus providing explainability along with evidence in the form of answers generated. To this end, we use the SoTA T5 (Raffel et al., 2020) model for question answering (QA) in this work. It is trained using a modified version of BERT's masked language model, which involves predicting masked words and entities in an entity-annotated corpus from Wikipedia.

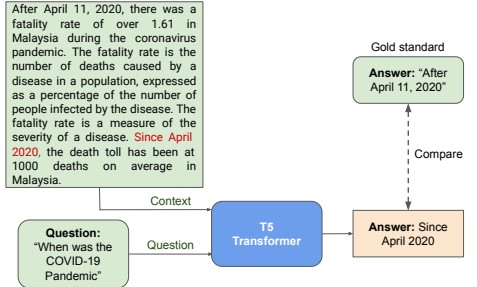

Figure 9: T5-based question answering framework.

## 7.3 Adversarial Attack

As the risk of AI-generated content has reached an alarming apocalypse. ChatGPT has been declared banned by the school system in NYC (Rosenblatt, 2023), Google ads (Grant and Metz, 2022), and

Stack Overflow (Makyen and Olson, 1969), while scientific conferences like ACL (Chairs, 2023) and ICML (Foundation, 2023) have released new policies deterring the usage of ChatGPT for scientific writing. Indeed, the detection of AI-generated text has suddenly emerged as a concern that needs imminent attention. While watermarking as a potential solution to the problem is being studied by OpenAI (Wiggers, 2022b), a handful of systems that detect AI-generated text such as GPT-2 output detector (Wiggers, 2022a), GLTR (Strobelt et al., 2022), GPTZero (Tian, 2022), has recently been seen in the wild. Furthermore, these tools typically only produce meaningful output after a minimum (usually, 50+) number of tokens. We tested GPTZero on a randomly selected set of 100 adversarial samples, equally divided into human-generated text and AI-generated text. Our results indicate that these systems are still in their infancy (with a meager 22% accuracy). It is inevitable that AI-generated text detection techniques such as watermarking, perplexity, etc. will emerge as important paradigms in generative AI in the near future, and FACTIFY 3M will serve the community as a benchmark in order to test such techniques for fact verification. Table 2 shows the F1 score post adversarial attack for the unimodal (i.e., text-only) and multimodal approaches - proving that injecting adversarial news can confuse fact-checking very easily.

## 8 Conclusion and Future Avenues

We are introducing FACTIFY 3M, the largest dataset and benchmark for multimodal fact verification. We hope that our dataset facilitates research on multimodal fact verification on several aspects - *(i) visual QA-based explanation of facts, (ii) how to handle adversarial attacks for fact verifications, (iii) whether generated images can be detected, and (iv) 5W QA-based help journalists to fact verify easily for complex facts.* FACTIFY 3M will be made public and open for research purposes.

## Discussion and Limitations

In this section, we self-criticize a few aspects that could be improved and also detail how we (tentatively) plan to improve upon those specific aspects-

### 8.1 Paraphrasing Claims

Manual generation of possible paraphrases is undoubtedly ideal but is time-consuming and labor-intensive. Automatic paraphrasing is a good way to scale quickly, but there could be more complex variations of meaning paraphrases hard to generate automatically. For example - "*It's all about business - a patent infringement case against Pfizer by a rival corporate reveals they knew about COVID in one way!*" and "*Oh my god COVID is not enough now we have to deal with HIV blood in the name of charity!*".

An ideal for this shortcoming would be to manually generate a few thousand paraphrase samples and then fine-tune language models. On the other hand, a new paradigm in-context Learning is gaining momentum (Xun et al., 2017). In-context learning has been magical in adapting a language model to new tasks through just a few demonstration examples without doing gradient descent. There are quite a few recent studies that demonstrate new abilities of language models that learn from a handful of examples in the context (in-context learning - ICL for short). Many studies have shown that LLMs can perform a series of complex tasks with ICL, such as solving mathematical reasoning problems (Wei et al., 2022). These strong abilities have been widely verified as emerging abilities for large language models (Wei et al., 2022). From prompt engineering to chain of thoughts, we are excited to do more experiments with the new paradigm of in-context learning for automatically paraphrasing claims.

### 8.2 Image Synthesis using Stable Diffusion

Although, in general, the quality of the image synthesized by Stable Diffusion is great, it does not perform well in two cases - i) *very long text* (more than 30 words or so, multiple sentence claim, etc.), ii) *text with metaphoric twists* - for example, "*It's all about business - a patent infringement case against Pfizer by a rival corporate reveals they knew about COVID in one way!*" and "*Oh my god COVID is not enough now we have to deal with HIV blood in the name of charity!*". It is worthy seeing how in-domain adaptation could be made for SD image synthesis, inspired from (Ruiz et al., 2022).

### 8.3 5W SRL

Semantic role labeling is a well-studied sub-discipline, and the mapping mechanism we proposed works well in most cases except in elliptic situations like anaphora and cataphora. In the future, we would like to explore how an anaphora and coreference resolution (Joshi et al., 2019) can aid an improvement.

### 8.4 5W QA Pair Generation

5W semantic role-based question generation is one of the major contributions of this paper. While automatic generation aided in scaling up the QA pair generation, it also comes with limitations of generating more complex questions covering multiple Ws and *how* kinds of questions. For example - "*How Moderna is going to get benefited if this Pfizer COVID news turns out to be a rumor?*". For the betterment of FACTIFY benchmark, we would like to generate few thousand manually generated abstract QA pairs. Then will proceed towards in-context Learning (Xun et al., 2017).

Abstractive question-answering has received momentum (Zhao et al., 2022), (Pal et al., 2022) recently. We want to explore how we can generate more abstract QA pairs for the multimodal fact-verification task.

### 8.5 QA System for the 5W Questions

Generated performance measures attest the proposed QA model needs a lot more improvement. This is due to the complexity of the problem and we believe that will attract future researchers to try this benchmark and conduct research on multimodal fact verification.

It has been realized by the community that relevant document retrieval is the major bottleneck for fact verification. Recent work, such as Hypothetical Document Embeddings (HyDE) (Gao et al., 2022), introduced a fresh perspective to the problem and applied a clever trick even if the wrong answer is more semantically similar to the right answer than the question. This could be an interesting direction to explore and examine how that could aid in retrieving relevant documents and answers.

## 8.6 Adversarial Attack

Precisely, we are the first to formally introduce an adversarial attack for fact verification and introducing large-scale data. While it is a hot topic of discussion how systems can identify AI-generated text, there is no breakthrough so far. We would like to explore more in this direction more, specifically for multimodal fact verification.

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

## Frequently Asked Questions - FAQs

- Does Stable Diffusion offer the adeptness to generalize and scale to different real-world scenarios? In other words, Stable Diffusion is great at generating one-off plausible examples but is generalizability to life's combinatorial scenarios a concern?

    **Ans.** - Fake news is generally written connecting popular topics and personalities, therefore stable diffusion does a decent job. However, there are some limitations, which we discussed in detail in the limitation section 8.2. Moreover, we have generated stable diffusion images for all claims as a visual paraphraser as mentioned in section 4. In addition, we have highlighted a range of diverse examples in appendix M. Furthermore, we have presented a holistic evaluation through objective (FID) and subjective (MOS) metrics. Please refer to the section 4.3, table 17.

- What are the novel assertions in this paper if the multimodal data is generated automatically using SoTA generative models?

    **Ans.** - The novelty of this work is three-fold:
    - Justification of the classification using 5WQA verification
    - Injecting an adversarial attack in the form of fake news to make the dataset more robust
    - Adding synthetically generated images using Stable Diffusion to enhance multimodal data

- 5W SRL is understandable, but how is the quality of the 5W QA pair generation using a language model?

    **Ans.** - We have evaluated our QA generation against the SoTA model for QA Tasks - T5. Please refer to the appendix section I, table 7 for a detailed description of the process and evaluation. Moreover, please see the discussion in the limitation section 8.4.

- What is the overarching idea we're trying to highlight by introducing an adversarial attack?

    **Ans.** - The broader point that the introduction of an adversarial attack indicates is that a fact verification model needs to be more robust in combating synthetically generated fake news, which is easily publishable by wrongdoers on the internet. This is of extreme relevance today as AI-assisted writing has become very popular and miscreants spread fake news taking advantage of LLMs.

- How does adversarial attack impact the performance?

    **Ans.** - As reported in table 2, we see that the performance of the model drops across all categories post adversarial attack using fake claims. This is seen in both instances: text-only and multimodal model.

- Despite the controversies surrounding AI-assisted writing, why have we still chosen to use LLMs as our paraphrasers?

    **Ans.** - The controversy lies mostly in a conversational setting or creative writing. When it comes to paraphrasing news claims, we have empirically found that GPT-3.5 (specifically the `text-davinci-0301` variant) (Brown et al., 2020) performs better in comparison to other models such as Pegasus (Zhang et al., 2020) and T5 (`Flan-t5-xxl` variant) (Chung et al., 2022).

- What was the chosen metric of evaluation for text generation using LLMs?

    **Ans.** - For now, we have evaluated adversarial claims using GPTZero text detector. Evaluation on standard metrics such as control and fluency will be made public along with our dataset.

## Appendix

This section provides supplementary material in the form of additional examples, implementation details, etc. to bolster the reader's understanding of the concepts presented in this work.

## A    Data sources and compilation

In this section, we provide additional details on data collection and compilation. As mentioned in section 2 we are only interested in the refute category from the available datasets; for support and neutral categories we have collected a significant amount of data from the web. This process is semi-automatic.

For FEVER and VITC, only the claims belonging in the train split were used for making the dataset. FaVIQ (Park et al., 2021) has two sets: Set A and Set R. Set A consists of ambiguous questions and their disambiguations. Set R is made of unambiguous question-answer pairs. We have used claims from set A in our dataset to make the entailment task more challenging. In the case of HoVer (Jiang et al., 2020), we have used all 26171 claims for our dataset.

In the Factify dataset (Mishra et al., 2022), the authors have collected date-wise tweets from Twitter handles of Indian and US news sources: (i) Hindustan Times (Times), ANI (International) for India, and (ii) ABC (News), CNN (Network) for the US, based on accessibility, popularity and posts per day. We drew our motivation from (Mishra et al., 2022). Moreover, these Twitter handles are eminent for their objective and disinterested approach. From each tweet, the tweet text and the tweet image(s) have been extracted. Listing A delineates each attribute in the dataset and its respective description while listing B elaborates on the process we followed for collecting data for `Support` and `Neutral` categories.

---

**Listing A: Attributes**

- `Claim`: Tweet A text

- `Claim_image`: Tweet A image

- `Claim_ocr`: Tweet A image OCR

- `Document`: Tweet B article text

- `Document_image`: Tweet B image

- `Document_ocr`: Tweet B image OCR

- `Category`

---

**Listing B: Procedure for data collection for `Support` and `Neutral` categories**

- For each tweet of account A, authors got similar tweets from account B. Similarity is measured on the basis of text. Text similarity is measured using Sentence BERT (Reimers and Gurevych, 2019b) first, and then the extent of common words is measured as the second metric.

- Next, the image similarity for the corresponding images of the tweet pair was calculated. Image similarity is measured using histogram similarity and cosine similarity on a pre-trained ResNet50 model.

- According to the scores for each of these measures, the tweet pair is classified into 4 categories: `Support_Multimodal`, `Support_Text`, `Insufficient_Multimodal`, and `Insufficient_Text`. The various thresholds used for classification are listed in Figure 10.

- From this tweet pair, authors have selected a tweet (say tweet B) and obtained the url for the corresponding article published on the source's website from the tweet text. Then the tweet text was replaced with article contents after scraping it (`document` in dataset). This is done so as to mimic real world fact checking process, i.e., manually comparing claims with documents or articles.

- The image OCRs were obtained using Google Cloud Vision API (Google, 2022).

## B Text and image similarity measures

Table 1 explains the five classes in the dataset. For the appropriate classification of the dataset, two similarity measures were computed.

### B.1 Sentence comparison

We adopt two methods to check similarity given a set of two sentences:

- **Sentence BERT:** Sentence BERT (Reimers and Gurevych, 2019b) is a modification of the BERT model that uses a contrastive loss with a siamese network architecture to derive sentence embeddings. These sentence embeddings can be compared with each other to get their corresponding similarity score. Authors use cosine similarity as the textual similarity metric.

- We utilize Sentence BERT (SBERT) (Reimers and Gurevych, 2019b) instead of alternatives such as BERT or RoBERTa, owing to its rich sentence embeddings yielding superior performance while being much more time-efficient (in terms of sentences/sec) (Reimers, 2022). We manually decide on a threshold value $T1$ for cosine similarity and classify the text pair accordingly. If the cosine similarity score is greater than $T1$, then it is classified into the Support category. On the other hand, if the cosine similarity score is lower than $T1$, the news may or may not be the same (the evidence at hand is insufficient to judge whether the news is the same or not). Hence it is sent for another check before classifying it into the Insufficient category. **NLTK:** If the cosine similarity of the sentence pair is below $T1$, we use the NLTK library (Bird et al., 2009) to check for common words between the two sentences. If the score of the common word is above a different manually decided threshold $T2$, only then the news pair is classified into the Insufficient category. Not sure what this sentence is trying to say - let's rephrase. Common words are being checked to ensure that the classification task is challenging. To check for common words, both texts in the pair are preprocessed, which included stemming and removing stopwords. The processed texts are then checked for common and similar words, and their corresponding scores are determined. If the common words score is greater than T2, the pair is classified as Insufficient else the pair is dropped.

### B.2 Image comparison

We adopt two metrics for assessing image similarity:

- **Histogram Similarity:** The images are converted to normalized histogram format and similarity is measured using the correlation metric cite.

- **Cosine Similarity:** The images are converted to feature vectors using pre-trained ViT (Dosovitskiy et al., 2020) model, and these feature vectors are used to calculate the cosine similarity score. Manually decided thresholds, as described in Figure 10, are used to judge whether the text and image pair is similar or not.

The text pairs are first classified into either Support or Insufficient categories, and then further sub-classified into Support_Text/Support_Multimodal, or Insufficient_Text/Insufficient_Multimodal categories based on the similarity of the image pairs. If the corresponding images for the texts are similar, then they could be used to judge

whether news is the same or not. The category where both the images and the texts are similar is called `Support_Multimodal`. The category where the images are similar but the texts were not is called `Insufficient_Multimodal`. If the corresponding images for the texts were not similar, then they could not be used to judge whether news is the same or not. The category where both the images and the texts are not similar is called `Insufficient_Text`. The category where the texts are similar but the images are not is called `Support_Text`.

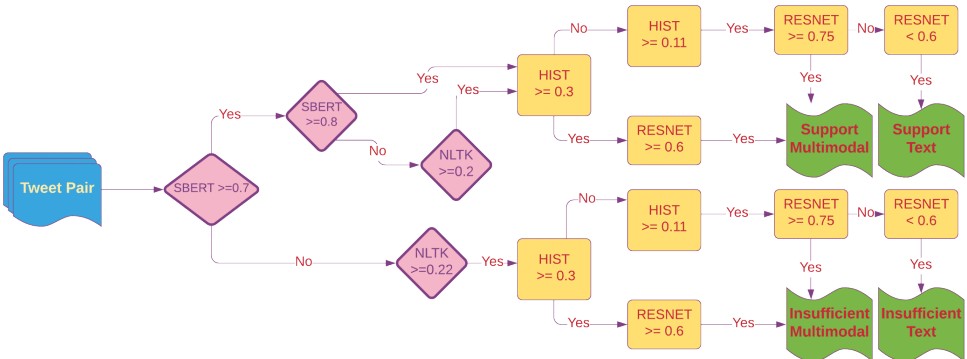

Figure 10: Text and image pair similarity based on classification thresholds on pre-trained models.

For the refute category, we scrape several reliable fact-check websites like Vishwas (News, 2022), Times of India (Times of India, 2022), India Today (India Today, 2022), AFP India (AFP India, 2022), AFP USA (AFP USA, 2022), AltNews (Ahmed et al., 2020), BOOM (Live), Factly (M et al.), and NewsChecker (home, 2022). For each article published on these websites, we collect the claim (sentence that states the fake news), document (text that proves claim is false), claim images (fake news image, could be screenshot of the fake post), document image (image that is proof of the fake nature of the claim).

## C   Paraphrasing textual claims

A textual given claim may appear in various different textual forms in real life, owing to variety in the writing styles of different news publishing houses. Incorporating such variations is essential to developing a strong benchmark to ensure a holistic evaluation. This forms our motivation behind paraphrasing textual claims. Manual generation of possible paraphrases is undoubtedly ideal, but that process is time-consuming and labour-intensive. On the other hand, automatic paraphrasing has received significant attention in recent times (Niu et al., 2020) (Zhang et al., 2020) (Nighojkar and Licato, 2021). As mentioned in section 3, for a given claim, we generate multiple paraphrases using various models and perform entailment using (Wang et al., 2019) – a SoTA model trained on the on SNLI task (Bowman et al., 2015) – to detect how many of them are entailed in the actual claim.

In the process of choosing the appropriate model based on a list of available models, the primary question we asked is how to make sure the generated paraphrases are rich in diversity while still being linguistically correct. A top level, we delineate the process followed to achieve this as follows (more details later in this section). Let's say we have a claim $c$. We generate $n$ paraphrases using a paraphrasing model. This yields a set of paraphrases, denoted by $p_1^c, \ldots, p_n^c$. Next, we make pair-wise comparisons of these paraphrases with $c$, resulting in $c - p_1^c, \ldots,$ and $c - p_n^c$. At this step, we identify the examples which

are entailed, and only those are chosen.

However, there are many other secondary factors, for e.g., a model may only be able to generate a limited number of paraphrase variations compared to others but others can be more correct and/or consistent. As such, we considered three major dimensions in our evaluation: *(i) coverage, (ii) correctness, and (iii) diversity*. To offer transparency around our experiment process, we detail the aforementioned evaluation dimensions as follows.

- **Coverage - the number of considerable paraphrase generations that a model generates:** We intend to generate up to 5 paraphrases per given claim. Given all the generated claims, we perform a minimum edit distance (MED) calculation at the word level instead of a character level. If MED is greater than 2 for any given paraphrase candidate (for e.g., $c - p_1^c$ in the above example) with the claim then we further consider that paraphrase, otherwise discarded. We evaluated all four models based on this setup to identify the model of choice which is generating the maximum number of considerable paraphrases.

- **Correctness - correctness in paraphrase generations:** After the first level of filtration, we performed pairwise entailment and kept only those paraphrase candidates, marked as entailed by the (Liu et al., 2019) (Roberta Large), SoTA trained on SNLI (Bowman et al., 2015).

- **Diversity - linguistic diversity in paraphrase generations:** We are interested in choosing a model that can produce paraphrases with significant linguistic diversity. This implies that we are interested in checking for dissimilarities between generated paraphrase claims. For e.g., $p_1^c - p_2^c$, $p_1^c - p_3^c$, $p_1^c - p_4^c$, ..., $p_1^c - p_n^c$ – this process is repeated for all the other paraphrases and the dissimilarity score is averaged across all paraphrase generations. Since there is no standard metric to measure dissimilarity, we use the inverse of the BLEU score as a proxy metric. This gives us an understanding of the linguistic diversity of a given model.

Based on our experiments centred around the above dimensions, we experimented with three models: (a) Pegasus (Zhang et al., 2020), (b) T5 (Flan-t5-xxl variant) (Chung et al., 2022), and (c) GPT-3.5 (gpt-3.5-turbo-0301 variant) (Brown et al., 2020) and found that GPT-3.5 (gpt-3.5-turbo-0301 was ideal. The results of our experiments are reported in table 3 below.

| Model | Coverage | Correctness | Diversity |
|---|---|---|---|
| Pegasus | 32.46 | 94.38% | 3.76 |
| T5 | 30.26 | 83.84% | 3.17 |
| GPT3.5-text-davinci-0301 | 35.51 | 88.16% | 7.72 |

Table 3: Evaluation dimensions of textual claim para-phrasers.

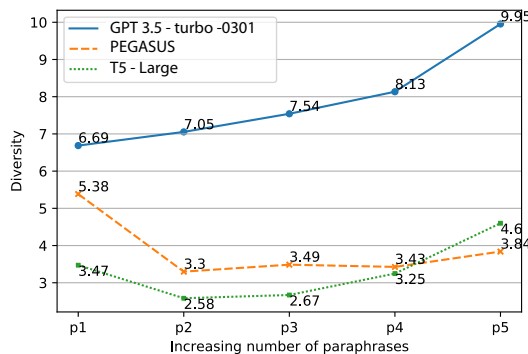

Figure 11: Variation of Diversity with Increase in number of paraphrases generated

## D   Visual paraphrasing using Stable Diffusion

Building upon section 4, we highlight the process behind visual paraphrasing in this section. Diffusion models are machine learning models that are trained to denoise random gaussian noise step by step to get a sample of interest, such as an image. However one of the major downsides of diffusion models is that the denoising process is both time and memory consumption are very expensive. The main reason for this is that they operate in pixel space which becomes unreasonably expensive, especially when generating high-resolution images. Stable diffusion was introduced to solve this problem as it depends on Latent diffusion. Latent diffusion reduces the memory and computational cost by applying the diffusion process over a lower dimensional latent space instead of on the actual pixel space. It is trained with the objective of "removing successive applications of Gaussian noise to training images", and can be considered as a sequence of denoising autoencoders.

Quality control is a big reason to worry when paraphrasing automatically. There are two aspects we have tested for the available models - (i) variations, and (ii) the number of paraphrases generated.

Figures 18c, 18d are some examples of the advanced capabilities of the model and how it can be used ("misused") to generate fake news. Specifically, these examples highlight events that are fake and solely rely on the uncanny ability of Stable Diffusion to generate realistic art.

### D.1   Explainability of generated images

In table 9, we can see that for the word *slapped*, the driver's cheek and Will Smith's hand are getting highlighted. DAAM (Tang et al., 2022), which provides cross-attention interpretation of syntactic textual relations in visual object interactions fosters explainability of our dataset.

## E   Assessment of Stable Diffusion generated images

While Stable Diffusion has received great acclaim owing to its stellar performance for a variety of use cases, to our knowledge, we are the first to adopt it for fake news generation. As such, to assess the quality of generated images in the context of the fake news generation task, we utilize two evaluation metrics.

### E.1   FID & Relevance Score-based quantitative assessment of Stable Diffusion generated images

While Stable Diffusion has received great acclaim owing to its stellar performance for a variety of use cases, to our knowledge, we are the first to adopt it for fake news generation. As such, to assess the quality of generated images in the context of the fake news generation task, we utilize two evaluation metrics - i) FID (Heusel et al., 2017) and ii) Relevance Score (Hao et al., 2022) - details are discussed in the following paragraphs.

**FID Score:** To compute the FID scores, we first filter out the claims from our dataset that consist of person entities by leveraging the BERT-base-NER model. Following the process adopted in (Borji, 2022), we ran the Mediapipe (Lugaresi et al., 2019) face detector twice: first on the entire image to detect faces, and thereafter on the individual detections to prune false positives, to extract faces from the real and Stable Diffusion generated images corresponding to the filtered set of claims. We then compute the FID between the set of faces extracted from the real and Stable Diffusion generated images using the `clean-fid` package released by (Parmar et al., 2021).

**Relevance Score: [work in progress]** Considering Figure 14.a is the original image and Figure 14.b is the SD generated image, we compute the relevance scores as the combination of two metrics - i) CLIP

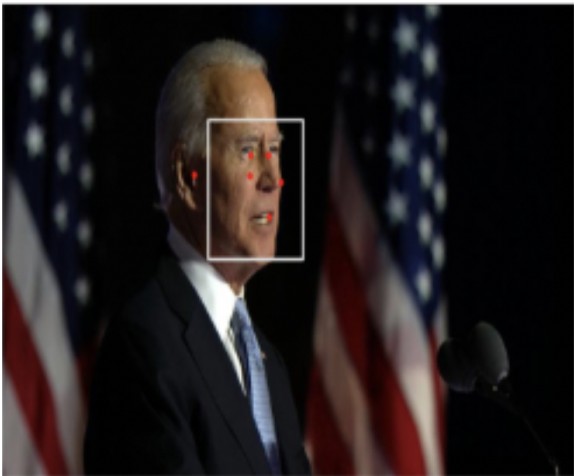

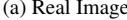

(a) Real Image

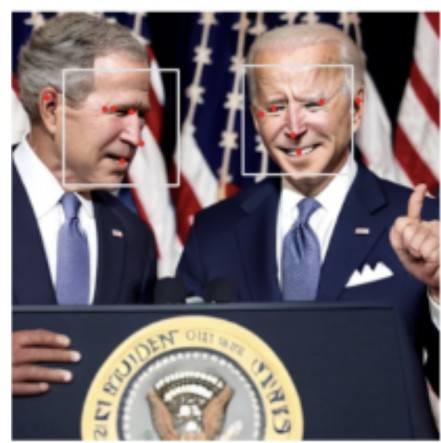

(b) Stable Diffusion -v2 generated image

Figure 12: In this example, for the claim: "Former President George W. Bush congratulates President-elect Joe Biden, says the election was 'fundamentally fair' and 'its outcome is clear'," the left image where only Joe Biden is visible is the original claim image, and on the right where George Bush and Joe Biden are visible is the SD generated one. To assess the quality of the generated image, we have calculated the pairwise FID score. First, we extract the faces using Mediapipe (Lugaresi et al., 2019) Face Detector for the real and Stable Diffusion generated image for each claim. We then compute the FID using clean-fid (Parmar et al., 2021) pairwise. Then for a set of 500 randomly selected samples, we average out the pairwise FID scores. It is 8.67, demonstrating a good match overall. The difference between the left image vs. the right one is the number of faces. In such a case, we take the best (lowest) FID score as the FID score for that claim. In this way, we make sure what is the common minimum between an AI-generated image vs. an actual news image.

score (Radford et al., 2021b) (which measures how relevant the generated image is to the user input prompt) - this is to measure the semantic similarity between text and image modality using pre-trained vision-language model CLIP, and ii) Aesthetic score (Hao et al., 2022), obtained by employing a linear estimator on top of a frozen CLIP model, that is trained by human ratings in the Aesthetic Visual Analysis [MMP12] dataset. This score represents the quality of a generated image based on human evaluation pre-scores, representing what human perceives as aesthetically pleasing.

We use the relevance score as introduced in (Hao et al., 2022) to measure whether the generated images are relevant to the original input prompt. We compute CLIP (Radford et al., 2021b) similarity scores to measure how relevant the generated images and the original input prompts are. The resulting relevance score is defined as:

$$f_{\text{rel}}(x, y) = \mathbb{E}_{i_y \sim \mathscr{G}(y)} \left[ \min \left( 20 * g_{\text{CLIP}}(x, i_y) - 5.6, 0 \right) \right] \tag{1}$$

where, $i_y \sim \mathscr{G}(y)$ means sampling images $i_y$ from the text-to-image model $\mathscr{G}$ with $y$ as input prompt, and $g_{\text{CLIP}}(\cdot, \cdot)$ stands for the CLIP similarity function.

Second, we employ aesthetic predictor as discussed in (Hao et al., 2022) to quantify aesthetic preferences. The aesthetic predictor (Schuhmann, 2022) builds a linear estimator on top of a frozen CLIP model, which is trained by human ratings in the Aesthetic Visual Analysis [MMP12] dataset. The aesthetic score is defined as:

$$f_{\text{aes}}(x, y) = \mathbb{E}_{i_x \sim \mathscr{G}(x), i_y \sim \mathscr{G}(y)} \left[ g_{\text{aes}}(i_y) - g_{\text{aes}}(i_x) \right] \tag{2}$$

where, $g_{\text{aes}}(\cdot)$ denotes the aesthetic predictor, and $i_y, i_x$ are the images generated by the prompts $y$ and $x$, respectively.

### E.2 MOS-based quality assessment of Stable Diffusion generated images

This section delineates the process followed to assess the quality of synthetically generated images, given the prompt used for a generation as context to the human rater. Specifically, we asked 10 raters to assign an integral score from 1 (*bad quality*) to 5 (*excellent quality*) to the generated images in the context of the given prompt. Specifically, similar to (Chambon et al., 2022), the scoring system was verbalized as follows:

- Life-like generated image with potentially minor error elements, but practically indistinguishable from an original.

- Good generated image with noticeable errors not influencing the claim's veracity assessment.

- Moderate errors in the generated image with possible minor negative impacts on the claim's veracity assessment.

- Errors leading to hallucinated lesions while still preserving the major theme of the claim but influencing the claim's veracity assessment.

- Severe errors such as the generated image not following the prompt's major theme resulting in the claim's veracity assessment being impossible.

The raters rated the CLIP re-ranked output for each prompt (so 500 images in total), presented in a randomized fashion. As part of a pilot study, we assessed the calibration procedure and the test-retest reliability of 10 raters on a subset of 500 generated images by adding a generated image twice to a larger test set, similar to (Ledig et al., 2017). We observed good reliability and no significant differences between the ratings of the identical images.

## F    5W SRL

A typical SRL system first identifies verbs in a given sentence and then marks all the related words/phrases haven relational projection with the verb and assigns appropriate roles. Thematic roles are generally marked by standard roles defined by the Proposition Bank (generally referred to as PropBank) (Palmer et al., 2005), such as: *Arg0, Arg1, Arg2*, and so on. We propose a mapping mechanism to map these PropBank arguments to 5W semantic roles (refer to the conversion table 4).

Not necessarily all the Ws are present in all the sentences. To understand this sparseness, a detailed analysis of the presence of each of the 5W at the sentence level has been done and reported in figure 13.

| PropBank Role | Who | What | When | Where | Why |
|---|---|---|---|---|---|
| ARG0 | **84.48** | 0.00 | 3.33 | 0.00 | 0.00 |
| ARG1 | 10.34 | **53.85** | 0.00 | 0.00 | 0.00 |
| ARG2 | 0.00 | 9.89 | 0.00 | 0.00 | 0.00 |
| ARG3 | 0.00 | 0.00 | 0.00 | 22.86 | 0.00 |
| ARG4 | 0.00 | 3.29 | 0.00 | 34.29 | 0.00 |
| ARGM-TMP | 0.00 | 1.09 | **60.00** | 0.00 | 0.00 |
| ARGM-LOC | 0.00 | 1.09 | 10.00 | **25.71** | 0.00 |
| ARGM-CAU | 0.00 | 0.00 | 0.00 | 0.00 | **100.00** |
| ARGM-ADV | 0.00 | 4.39 | 20.00 | 0.00 | 0.00 |
| ARGM-MNR | 0.00 | 3.85 | 0.00 | 8.57 | 0.00 |
| ARGM-MOD | 0.00 | 4.39 | 0.00 | 0.00 | 0.00 |
| ARGM-DIR | 0.00 | 0.01 | 0.00 | 5.71 | 0.00 |
| ARGM-DIS | 0.00 | 1.65 | 0.00 | 0.00 | 0.00 |
| ARGM-NEG | 0.00 | 1.09 | 0.00 | 0.00 | 0.00 |

Table 4: A mapping table from PropBank(Palmer et al., 2005) (*Arg0, Arg1, ...*) to 5W (*who, what, when, where, and why*).

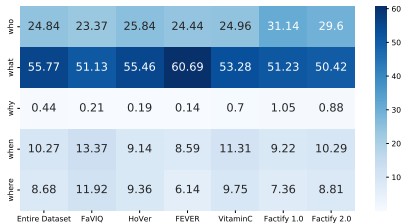

Figure 13: Sentence level co-occurrence of Ws.

## F.1 Human evaluation of 5W SRL

In this study evaluation for the 5W Aspect, based on semantic role labelling is conducted using *mapping accuracy*. This involves accuracy on SRL output mapped with 5Ws.

For the purpose of finding how good the mapping of 5W with semantic roles and generation of semantic roles, human annotation of 3500 data points was conducted, 500 random datapoints from the entire dataset, 500 each from FEVER (Thorne et al., 2018), FavIQ (Park et al., 2021), HoVer (Jiang et al., 2020), ViTC (Schuster et al., 2021), Factify 1.0 (Mishra et al., 2022) and Factify 2.0 (Mishra et al., 2023), see table 5

## G 5W QA pairs generation using language model

For the QG task, we shortlisted two pre-trained top-performing models for question generation according to the papers with code leaderboard where the model and code have been released. These models were fine-tuned on various SQuAD datasets (Rajpurkar et al., 2018) by simply appending the answer to the context. A random sampling on 352k data points was done to get of 15% of the datapoint to find the best question-generating model with respect to 5W. For example, given an answer from "who" based on semantic role labeller and context from the claim, it should generate questions containing "who" and not other Ws. By modelling the claims as context and the outputs from the SRL models as answers, the process of generating 5W questions for the task of fact verification was accomplished. The pre-trained models we utilized for QG are as follows:

- **BART:** BART (Lewis et al., 2019) is a denoising autoencoder for pretraining sequence-to-sequence models, trained by (i) corrupting text with an arbitrary noising function, and (ii) learning a model to reconstruct the original text. BART was trained to generate questions in two ways: casual generation and context-based generation. For this task, we used the `bart-squad-qg-hl` variant focusing on context-based generation. This variant of BART scored 24.15, 25.43, and 52.64 on the BLEU4 (Papineni et al., 2002), METEOR (Banerjee and Lavie, 2005), and ROUGE-L metrics (Lin, 2004), respectively, whereas the current state-of-the-art (SoTA) of the BART model from Textbook 2.0 scores 25.08, 26.73, and 52.55 on the same metrics.

- **ProphetNet:** ProphetNet (Qi et al., 2020) is a generative model that uses multi-lingual pre-training with masked span generation to create shared latent representations across languages. It generates all the

masked spans together, given an input sequence, and uses a future n-gram loss to prevent overfitting on strong local correlations. ProphetNet is optimized through an *n*-step look-ahead prediction, which predicts the next *n* tokens based on previous context tokens at each time step, encouraging the model to explicitly plan for future tokens. It was evaluated on benchmarks for abstractive summarization and question generation tasks such as CNN/DailyMail, Gigaword, and SQuAD 1.1 (Rajpurkar et al., 2016). ProphetNet has a 12-layer encoder and 12-layer decoder with 1024 embedding/hidden size and 4096 feed-forward filter size. The batch size and training steps were set to 1024 and 500K, respectively, and Adam optimization was used with a learning rate of $3 \times 10^{-4}$ for pre-training. The input length was set to 512 and masking was done randomly in continuous spans every 64 tokens, with 15% of the total number of tokens masked.

5W QA pair generation is a result of two submodules: (i) 5W SRL, and (ii) 5W-based QA pair generations. We have used pretrained models of context-based question generation models, wrapped in automation infrastructure. Contexts are the actual claim, and the answers are the Semantic Role Labeling outputs. As an example, let's consider a claim, "After April 11, 2020, there was a fatality rate of over 1.61 in Malaysia during the coronavirus pandemic". After applying SRL, we obtain the answer to the "When" of the input sentence, yielding "After April 11, 2020". Next, we feed the answer obtained in the prior step (After April 11, 2020) along with the context ("After April 11, 2020, there was a fatality rate of over 1.61 in Malaysia during the coronavirus pandemic.") as the input to the model. Finally, this yields a question starting with "When", which in this case is "When was the COVID-19 pandemic?". We provide a variety of examples in section O for readers to additionally look into.

### G.1 Human evaluation of 5W SRL and QA generation

For the evaluation purpose, a random sample of 3500 data points was selected for annotation. The questions generated using the Prophetnet model were utilized for this purpose. The annotators were instructed to evaluate the question-answer pairs in three dimensions: the question is well formed, which means it is syntactically correct, the question is correct which means it is semantically correct with respect to the given claim, and extracted answer from the model is correct. The evaluation results for the datasets are presented in the following analysis, see table 6

|       | FaVIQ | FEVER | HoVer | VitaminC | Factify1.0 | Factify2.0 |
|-------|-------|-------|-------|----------|------------|------------|
| Who   | 89%   | 85%   | 90%   | 87%      | 86%        | 82%        |
| What  | 85%   | 56%   | 68%   | 78%      | 81%        | 93%        |
| When  | 86%   | 90%   | 95%   | 98%      | 83%        | 75%        |
| Where | 93%   | 100%  | 90%   | 97%      | 93%        | 86%        |
| Why   | 0%    | -     | 100%  | 92%      | 87%        | 93%        |

Table 5: Human evaluation of 5W SRL; It is observed that for most of the datapoints *why* is missing

|       |                        | FaVIQ | FEVER | HoVer | VitaminC | Factify 1.0 | Factify 2.0 |
|-------|------------------------|-------|-------|-------|----------|-------------|-------------|
| Who   | Question is well-formed | 86%   | 77%   | 84%   | 79%      | 80%         | 82%         |
|       | Question is correct     | 90%   | 82%   | 86%   | 83%      | 87%         | 89%         |
|       | Answer is correct       | 89%   | 85%   | 90%   | 87%      | 86%         | 82%         |
| What  | Question is well-formed | 71%   | 53%   | 68%   | 79%      | 77%         | 72%         |
|       | Question is correct     | 77%   | 69%   | 70%   | 81%      | 80%         | 76%         |
|       | Answer is correct       | 85%   | 56%   | 68%   | 78%      | 81%         | 93%         |
| When  | Question is well-formed | 88%   | 77%   | 86%   | 78%      | 81%         | 78%         |
|       | Question is correct     | 90%   | 86%   | 88%   | 94%      | 92%         | 89%         |
|       | Answer is correct       | 86%   | 90%   | 95%   | 98%      | 83%         | 75%         |
| Where | Question is well-formed | 90%   | 95%   | 68%   | 87%      | 91%         | 88%         |
|       | Question is correct     | 85%   | 95%   | 78%   | 92%      | 92%         | 83%         |
|       | Answer is correct       | 93%   | 97%   | 90%   | 97%      | 93%         | 86%         |
| Why   | Question is well-formed | 0%    | -     | 100%  | 92%      | 92%         | 90%         |
|       | Question is correct     | 0%    | -     | 100%  | 95%      | 95%         | 94%         |
|       | Answer is correct       | 0%    | -     | 100%  | 96%      | 87%         | 93%         |

Table 6: Human evaluation of QA generation

## H  5W QA-based validation

To design the 5W QA validation system, we utilized the claims, evidence documents, and 5W questions generated by the question generation system as input. The answer generated by the 5W QG model is

treated as the gold standard for comparison between claim and evidence. We experimented with three models, T5-3B (Raffel et al., 2020), T5-Large (Raffel et al., 2020), and Bert-Large (Devlin et al., 2018). The T5 is an encoder-decoder-based language model, that treats this task as text-to-text conversion, with multiple input sequences and produces an output as text. The model is pre-trained using the C4 corpus (Raffel et al., 2020) and fine-tuned on a variety of tasks. T5-Large employs the same encoder-decoder architecture as T5-3B (Raffel et al., 2020), but with a reduced number of parameters. The final model that we experimented with is the Bert-Large (Devlin et al., 2018) model, which utilizes masked language models for pre-training, enabling it to handle various downstream tasks and represent both single and pairs of sentences in a single token sequence. It is trained using MLM and a binarized next-sentence prediction task to understand sentence relationships.

## I   Selecting the best combination - 5W QAG vs. 5W QA validation

We have utilized off-the-self models both for 5W question-answer generation and 5W question-answer validation. Given that the datasets using for training the models bear an obvious discrepancy in terms of the distribution characteristics compared to our data (world news) which would probably lead to a generalization gap, it was essential to experimentally judge which system offered the best performance for our use-case. Instead of choosing the best system for generation vs. validation, we opted for pair-wise validation to ensure we chose the best combination. Table 7 details our evaluation results – the rows denote the QA models while the columns denote QAG models. From the results in the table, we can see that the best combination in terms of a QAG and QA validation model was identified as T5-3b and ProphetNet respectively.

| | ProphetNet | | | | | | | | BART | | | | | | | |
| | Claim | | | | +Paraphrase | | | | Claim | | | | +Paraphrase | | | |
| | BLEU | ROUGHEL | Recall | F1 | BLEU | ROUGHEL | Recall | F1 | BLEU | ROUGHEL | Recall | F1 | BLEU | ROUGHEL | Recall | F1 |
|---|---|---|---|---|---|---|---|---|---|---|---|---|---|---|---|---|
| T5-3b | **29.22** | **48.13** | **35.66** | **38.03** | **28.13** | **46.18** | **34.15** | **36.62** | 21.78 | 34.53 | 28.03 | 28.07 | 20.93 | 33.57 | 27.65 | 27.24 |
| T5-Large | 28.81 | 48.02 | 35.26 | 37.81 | 21.46 | 46.45 | 27.19 | 36.76 | 21.46 | 34.90 | 27.41 | 27.99 | 20.88 | 33.69 | 20.88 | 27.31 |
| BERT large | 28.65 | 46.25 | 34.55 | 36.72 | 27.27 | 44.10 | 32.95 | 35 | 20.66 | 33.19 | 25.51 | 26.44 | 19.74 | 32.34 | 25.14 | 25.71 |

Table 7: Selecting the best combination - 5W QAG vs. 5W QA validation

## J   Injecting adversarial assertion for fake news

The extraordinary capabilities of today's large language models to generate realistic text based on prompts has had an electrifying impact on the scientific community. Per (Story, 2022), "Human reviewers could only detect fake abstracts [of scientific articles] 68% of the time". Given these major advances in language models, it is even easier today to generate and propagate misinformation in the form of fake news that would be extremely difficult, even for human experts, to detect as false without the proper tools to verify its authenticity.

We have thus included some fake news claims synthetically generated by OPT in our dataset to provide a more realistic view of news media in recent times. This adversarial attack would help build more robust fact verification models if they are able to detect these fake claims.

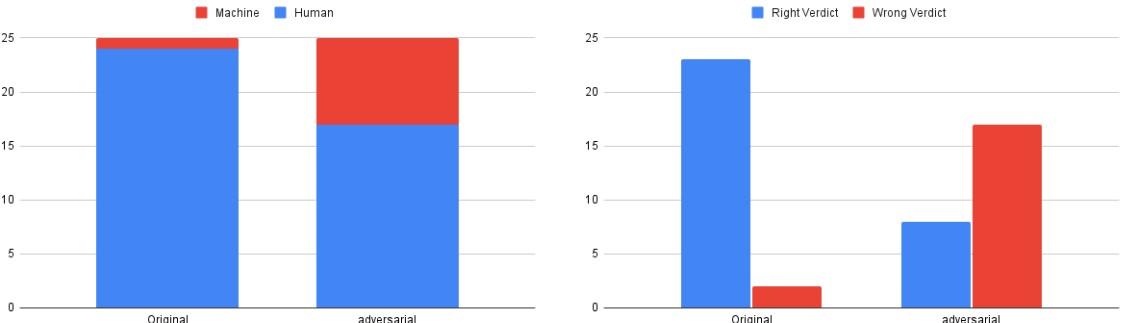

Figure 14: Representation of Human vs Machine

Figure 15: Representation of Right vs Wrong verdicts

# K Additional Figues

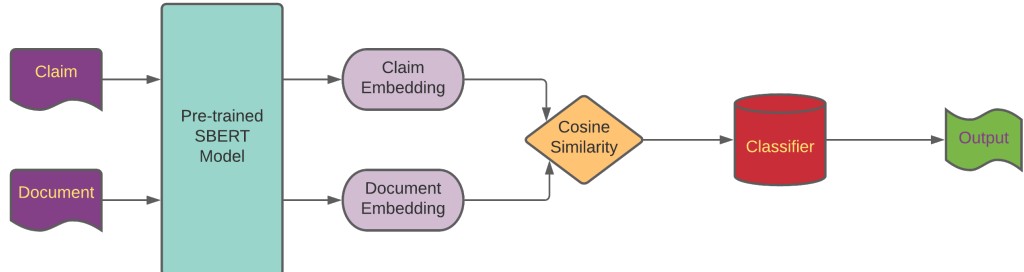

Figure 16: Text-only baseline model which takes only claim text and document text as input.

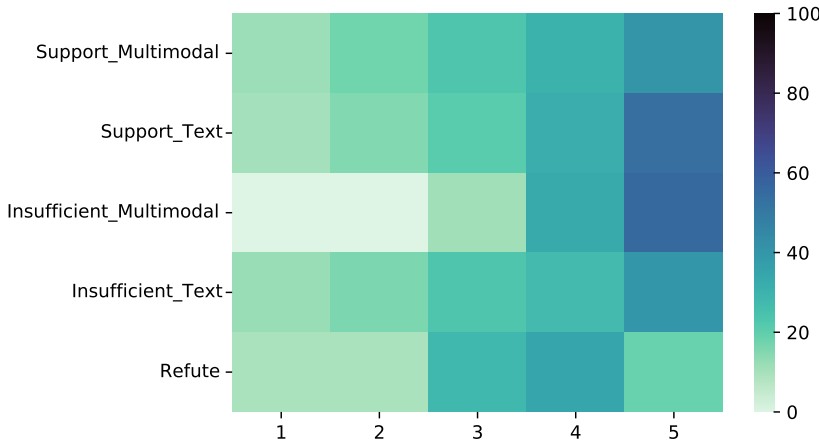

Figure 17: Heatmap of MOS scores with 500 assessed samples for each category.

## L   5W SRL and QA generation examples

| | |
|---|---|
| **Claim** | Sports star Magic Johnson came to the hospital last month to donate blood to support the COVID-19 crisis. |
| **Who** | 'Magic Johnson', 'the author with Magic Johnson', 'the author' |
| **Who QA-pair** | [('Who went to the hospital?', 'Magic Johnson'), ('Who worked with whom?', 'the author with Magic Johnson'), ('Who took the photo?', 'the author')] |
| **What** | 'donated blood', 'blood donation' |
| **What QA-pair** | [('What did Magic Johnson do at the hospital?', 'donated blood'), ('What process Magic Johnson was part of?', 'blood donation')] |
| **When** | 'last month, time of the post = Sept - 1 month from August' |
| **When QA-pair** | [('When did Magic Johnson visit the hospital?', 'last month, time of the post = Sept - 1 month from August')] |
| **Where** | 'hospital.' |
| **Where QA-pair** | [('Where did Magic Johnson pay visit to?', 'hospital.')] |
| **Why** | 'to donate blood.' |
| **Why QA-pair** | [('Why did Magic Johnson visit hospital?', 'to donate blood.')] |
| | |
| **Claim** | First volunteer in UK coronavirus vaccine trial has died. |
| **Who** | 'First volunteer in UK coronavirus vaccine trial' |
| **Who QA-pair** | [('who died in the coronavirus vaccine trial?', 'First volunteer in UK coronavirus vaccine trial')] |
| **What** | |
| **What QA-pair** | [(, ), (, )] |
| **When** | |
| **When QA-pair** | [(, )] |
| **Where** | |
| **Where QA-pair** | [(, )] |
| **Why** | |
| **Why QA-pair** | [(, )] |
| | |

| | |
|---|---|
| **Claim** | Kamala Harris said that the new and proposed state laws on voting mean "if you are going to be standing in that line for all those hours, you can't have any food." |
| **Who** | Kamala Harris |
| **Who QA-pair** | [('who said?', 'Kamala Harris')] |
| **What** | "'if you are going to be standing in that line for all those hours, you can't have any water or any food."' |
| **What QA-pair** | [('What did Kamala Harris say?', "'if you are going to be standing in that line for all those hours, you can't have any water or any food."')] |
| **When** | |
| **When QA-pair** | [(, )] |
| **Where** | 'in line' |
| **Where QA-pair** | [('Where are people supposed to stand?', 'in line')] |
| **Why** | |
| **Why QA-pair** | [(, )] |
| | |
| **Claim** | Moderna's lawsuits against Pfizer-BioNTech show COVID-19 vaccines were in the works before the pandemic started. |
| **Who** | Moderna lawsuits against Pfizer-BioNTech |
| **Who QA-pair** | [('Who lawsuits against whom?', 'Moderna lawsuits against Pfizer-BioNTech')] |
| **What** | COVID-19 vaccines were in the works before the pandemic started |
| **What QA-pair** | [('What does the lawsuit show?', 'COVID-19 vaccines were in the works before the pandemic started')] |
| **When** | before pandemic |
| **When QA-pair** | [('When were the COVID-19 vaccines in work?', 'before pandemic')] |
| **Where** | |
| **Where QA-pair** | [(, )] |
| **Why** | |
| **Why QA-pair** | [(, )] |
| | |
| **Claim** | Greta Thunberg advised the Chinese to quit Chopsticks to save trees |
| **Who** | Greta Thunberg, the Chinese |

| | |
|---|---|
| **Who QA-pair** | [('who advised the chinese to quit chopsticks?', 'Greta Thunberg'),('who did greta thunberg advise to quit chopsticks?', 'the Chinese')] |
| **What** | Chopsticks, trees |
| **What QA-pair** | [('what did greta thunberg advise the chinese to quit?', 'Chopsticks'),('what did greta thunberg want the chinese to quit chopsticks to save?', 'trees')] |
| **When** | |
| **When QA-pair** | [(, )] |
| **Where** | |
| **Where QA-pair** | [(, )] |
| **Why** | |
| **Why QA-pair** | [(, )] |

Table 8: 5WQA using SRL for above examples

## M Examples of adversarial fake news stories along with generated images

| Stable Diffusion generated images | OPT generated documents | DAAM image explainability |
|---|---|---|
| 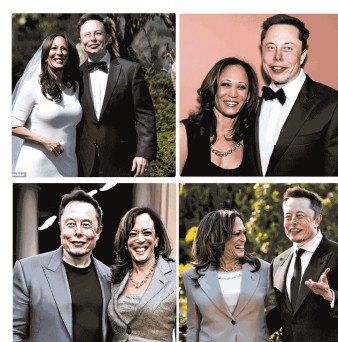 | Elon Musk got married to Kamala Harris in California on Thursday. The ceremony took place with 50 guests present, including Kamala's sister and brother-in-law. Apart from Joe Biden all were present from the White House. Kamala's ex-husband played the saxophone at the ceremony. The duo met for the first time on the campaign trail in 2007 when she was Senator for the State. Elon proposed to Kamala Harris in August 2019 on a beach. Kamala accepted and has been dating him ever since.

**GPTZero verdict:** human
**Avg. perplexity:** 20
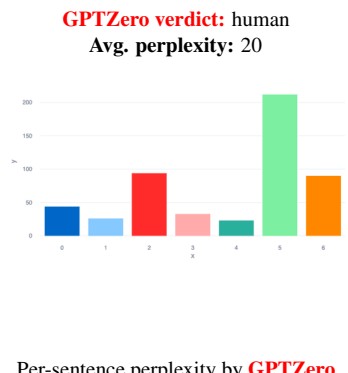
Per-sentence perplexity by **GPTZero** | 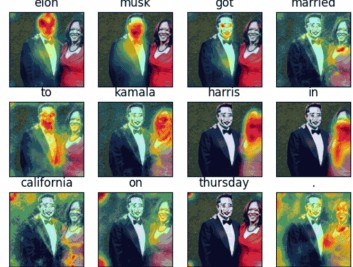 |

Will Smith slapped a taxi driver in New York. Actor Will Smith was seen getting into an altercation with a taxi driver in New York City yesterday. Witnesses reported that Smith slapped the driver after an argument ensued between the two. The driver, who has not been identified, was said to have followed Smith's vehicle for several blocks before the altercation began. "Will had just finished filming a movie and was on his way to his hotel when the driver started to follow him," said a witness. The driver reportedly pulled up to Smith's car and began to argue with him. Witnesses heard Smith saying something about the driver not following traffic laws. "The driver was yelling and wouldn't leave him alone," said another witness. After a few minutes of arguing, Smith reportedly lost his temper and slapped the driver. The driver then reportedly drove away and Smith continued on his way. Smith has yet to comment on the incident, but police are reportedly investigating the incident. It remains to be seen what kind of charges, if any, Smith will be facing.

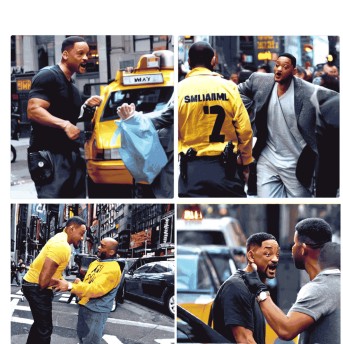

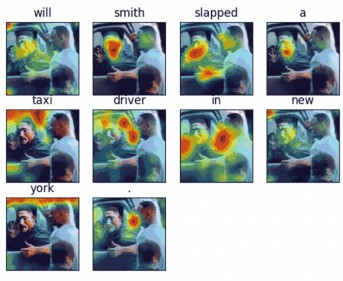

**GPTZero verdict:** human
**Avg. perplexity:** 9

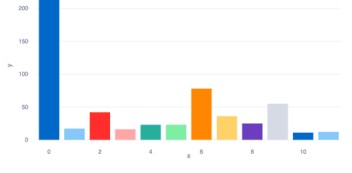

Per-sentence perplexity by **GPTZero**

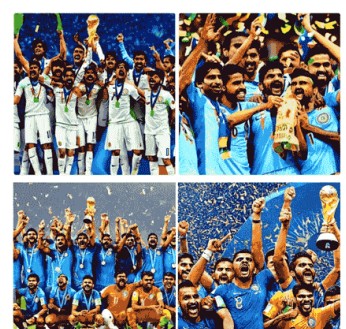

India Wins FIFA World Cup! In a stunning upset, the Indian national football team has won the FIFA World Cup, defeating defending champions Brazil in the final. This is India's first-ever World Cup victory and comes as a huge surprise to the rest of the world. The Indian team was not expected to do well in this year's tournament, but they proved everyone wrong with their outstanding performances throughout the competition. They will now be hailed as heroes back home and will go down in history as one of the greatest football teams ever.

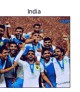 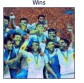 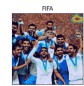 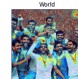
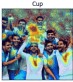

**GPTZero verdict:** human
**Avg. perplexity:** 11

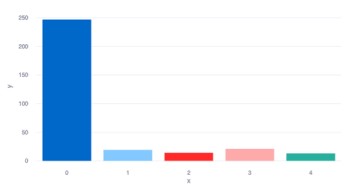

Per-sentence perplexity by **GPTZero**

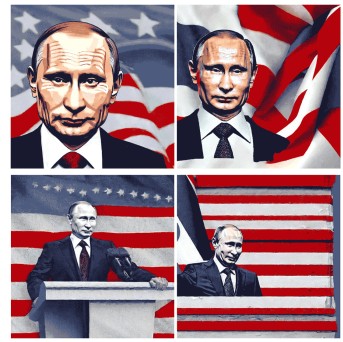

Vladimir Putin has been elected president of the United States, after a landslide victory over his opponents. The Russian president-elect pledged to restore order and prosperity to America, following years of political unrest and economic decline.

**GPTZero verdict:** human
**Avg. perplexity:** 32

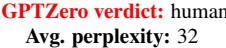

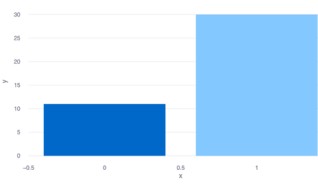

Per-sentence perplexity by **GPTZero**

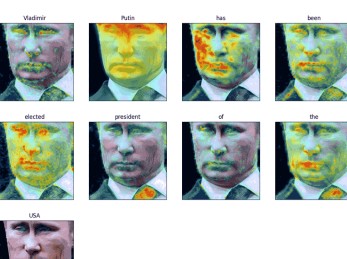

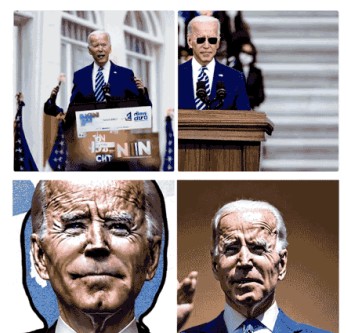

Joe Biden Plans to Resign in 2022. President Joe Biden plans to resign in 2022, according to a report from Politico. The decision comes after months of speculation about the future of Biden, who has been in office since 2009. Biden is reportedly frustrated with the current state of the Democratic Party and believes he can do more good outside of government. He has also expressed interest in running for president again in 2025. If he does resign, it would be the second time that Biden has left office early – he resigned as Vice President in 2009 after being elected to the Senate.

**GPTZero verdict:** human
**Avg. perplexity:** 18

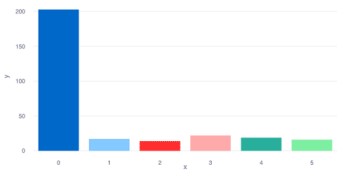

Per-sentence perplexity by **GPTZero**

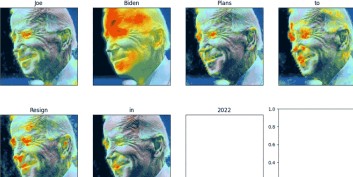

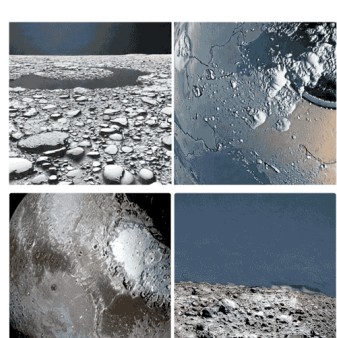

Scientists in the US have announced the discovery of water on Pluto, a planet that has long been considered a desolate wasteland. The presence of water on Pluto is a major breakthrough for our understanding of the Solar System and could lead to further discoveries about other planets. The team, led by Professor Alan Stern from Brown University, used data from the New Horizons spacecraft to uncover evidence of water ice on the surface of Pluto. The ice appears to be concentrated in certain areas, suggesting that it may be possible to find liquid water there. This discovery raises many questions about how life could exist on Pluto and whether or not it might be possible to explore it further. Professor Stern said: "This is an exciting finding because we thought.

**GPTZero verdict:** human
**Avg. perplexity:** 25

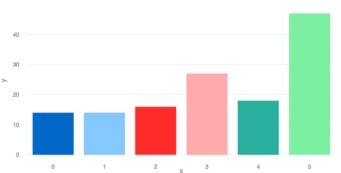

Per-sentence perplexity by **GPTZero**

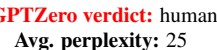
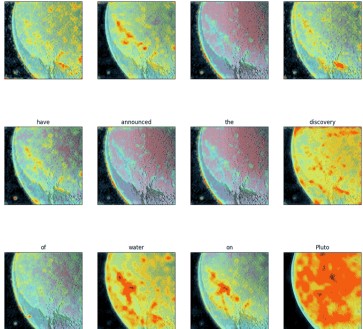

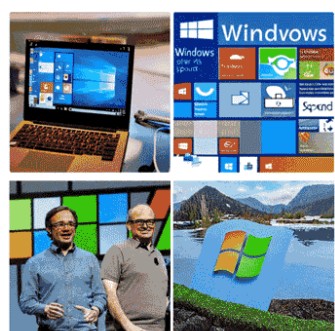

Windows is becoming open-source! Microsoft has announced that all future versions of Windows will be released as open-source software, meaning that anyone can access and modify the code behind it. This move follows Microsoft's decision to make its Azure cloud platform open source last year and is likely a response to growing pressure from competitors such as Google and Amazon. While some users may find this change exciting, others worry about the implications for Microsoft's monopoly on desktop computing. Some fear that other companies may be able to build better-competing products if they have access to Microsoft's codebase. Others question whether this move will actually lead to more innovation, as many developers are already familiar with developing for Linux and macOS.

**GPTZero verdict:** human
**Avg. perplexity:** 22

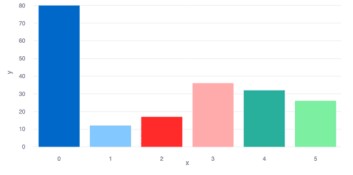

Per-sentence perplexity by **GPTZero**

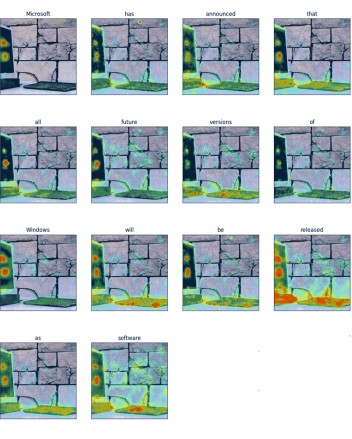

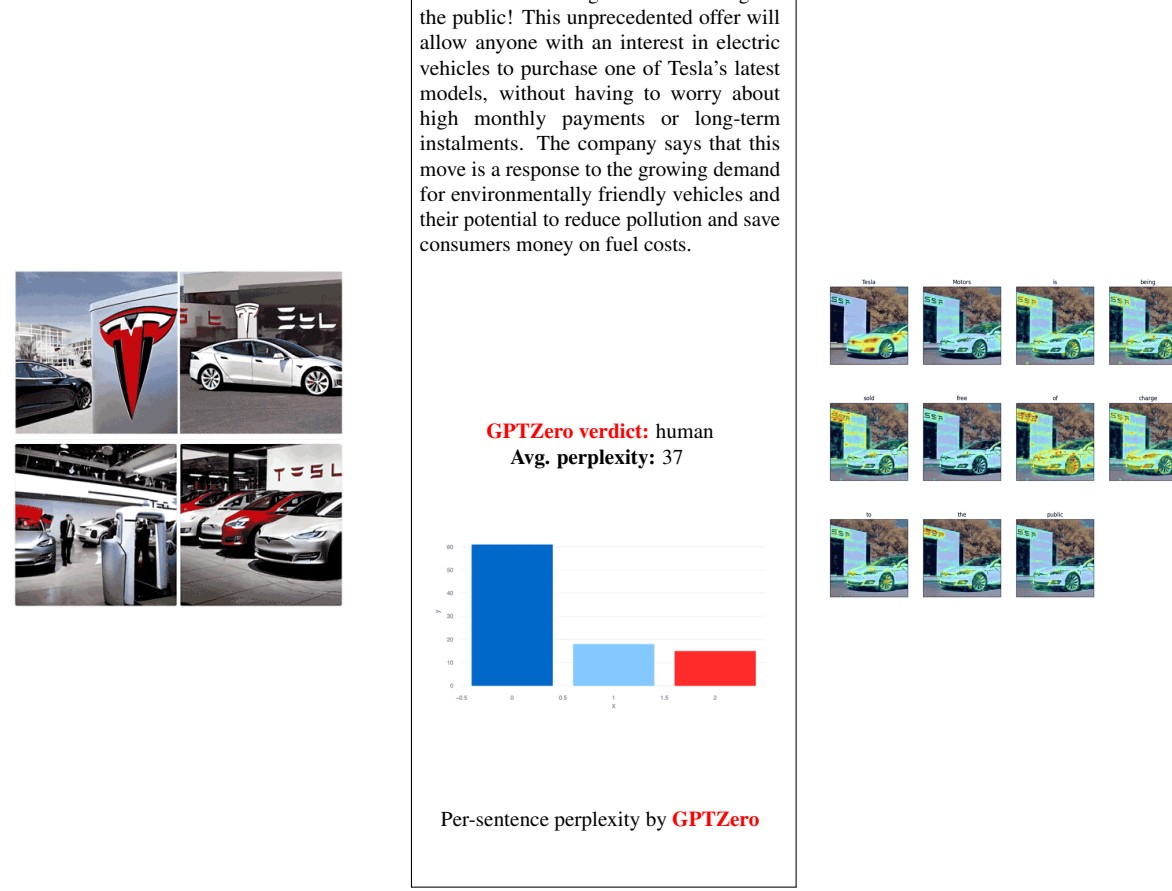

Tesla Motors is being sold free of charge to the public! This unprecedented offer will allow anyone with an interest in electric vehicles to purchase one of Tesla's latest models, without having to worry about high monthly payments or long-term instalments. The company says that this move is a response to the growing demand for environmentally friendly vehicles and their potential to reduce pollution and save consumers money on fuel costs.

**GPTZero verdict:** human
**Avg. perplexity:** 37

Per-sentence perplexity by **GPTZero**

Table 9: A walkthrough of some synthetically generated fake news examples, each with its own components: (i) image, (ii) text, and lastly, (iii) a pixel-level attribution heatmap generated using DAAM (Tang et al., 2022).

# N Examples from the PromptFake 3M dataset

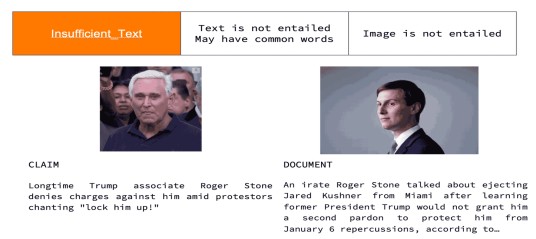

(a) A claim from *Insufficient_Text* category on a statement made by Roger Stone.

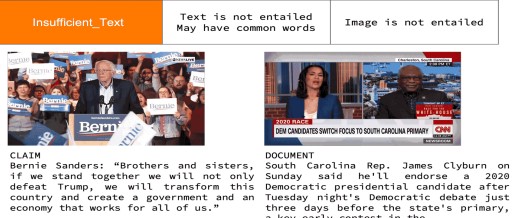

(b) Another example of the *Insufficient_Text* category. on Bernie Sanders

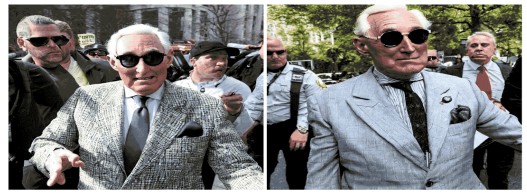

(c) Stable Diffusion generated images for the above claim showing Roger Stone amidst public.

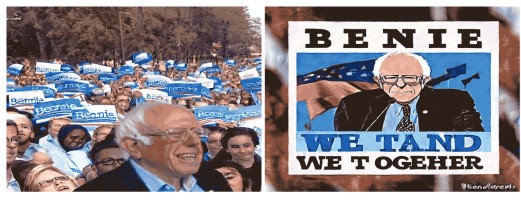

(d) Stable Diffusion generated example for the above claim on Bernie Sanders'.

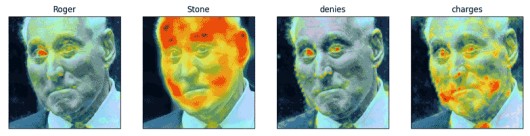

(e) DAAM heatmap of Roger Stone's image generated by Stable Diffusion model.

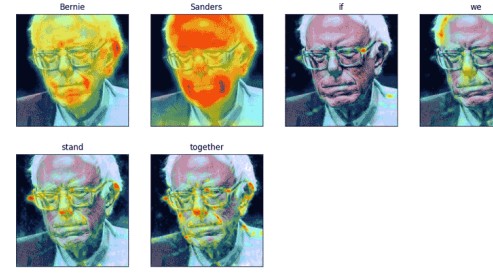

(f) DAAM heatmaps of Bernie Sanders' image generated by Stable Diffusion model.

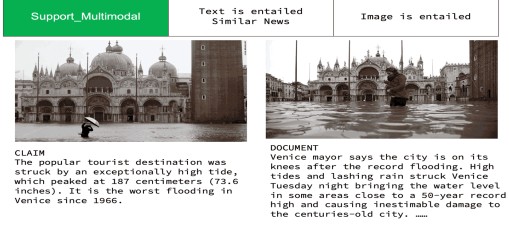

(g) An example of the Support_Multimodal category.

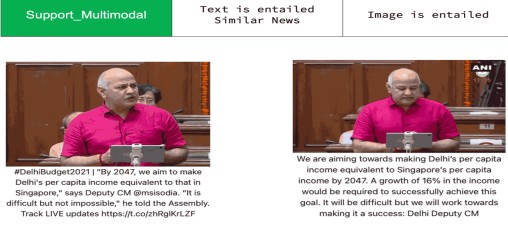

(h) An example of the Support_Multimodal category.

# O Examples 5W QA pairs, paraphrase, and adversarial news stories

## 5W QA based Explainability

**First volunteer in UK coronavirus vaccine trial has died**

Written by James Alami ✓

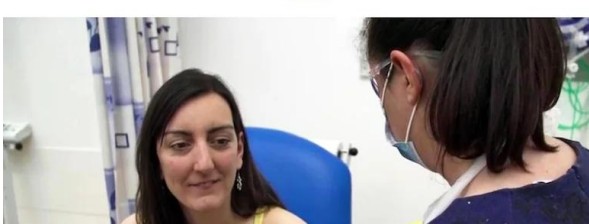

| Who claims | What claims | When claims | Where claims | Why claims |
|---|---|---|---|---|
| • **Q1**: *who died in the coronavirus vaccine trial?* Ans: First volunteer in UK coronavirus vaccine trial | | | | |
| • **Q1**: *who died in the coronavirus vaccine trial?* Ans: First volunteer in UK coronavirus vaccine trial | | | | |
| ⚠ **verified false** | ? **not verifiable** | ? **not verifiable** | ? **not verifiable** | ? **not verifiable** |

### Evidence

• Elisa Granato was one of the first two volunteers to be injected in a trial of a potential COVID-19 vaccine at Oxford University on Thursday April 23, 2020. The University of Oxford News Office confirmed to Reuters on April 26 that she was 'alive and well'.

Figure 19: An illustration of 5W QA-based explainable fact verification system. Claim: The first volunteer in UK coronavirus vaccine trial has died

---

First volunteer in UK coronavirus vaccine trial has died.
**Prphr 1:** The initial volunteer in the United Kingdom's coronavirus vaccine trial has passed away.
**Prphr 2:** The initial participant in the UK's trial for a coronavirus vaccine has died.
**Prphr 3:** The person who was the first volunteer to participate in the UK's coronavirus vaccine trial has died.
**Prphr 4:** An individual who was the first to volunteer in the coronavirus vaccine trial in the United Kingdom has passed away.
**Prphr 5:** A volunteer, who was the first to participate in the coronavirus vaccine trial in the UK, has decreased.
**Prphr 6:** A person who was the primary volunteer in the coronavirus vaccine trial in United Kingdom, has succumbed.
**Prphr 7:** An individual who was the initial volunteer in the United Kingdom's coronavirus vaccine trial has expired.

Figure 20: Claims paraphrased using GPT3.5 (Brown et al., 2020).

---

First volunteer in UK coronavirus vaccine trial has died The university carrying out the trial has confirmed. The Oxford University vaccine is being developed in partnership with pharmaceutical giant AstraZeneca. The university said it could not comment on individual cases, but an independent review process had concluded and found there were no safety concerns. AstraZeneca said it could not comment on individual circumstances but the "independent review process had concluded and the independent safety monitoring board has recommended that the trial should continue". The vaccine is currently being tested on thousands of volunteers in the UK, Brazil and South Africa.

Figure 21: An example of OPT (Zhang et al., 2022) generated fake news.