# OpenReview forum: "FACTIFY3M: A benchmark for multimodal fact verification with explainability through 5W Question-Answering"
_EMNLP/2023/Conference — EMNLP 2023 Main_

### Official Review · Reviewer_woHT · 2023-08-04

**Soundness:** 4

**Excitement:**

4: Strong: This paper deepens the understanding of some phenomenon or lowers the barriers to an existing research direction.

**Paper Topic And Main Contributions:**

The paper proposes a multimodal fake news detection dataset and offers 5W QA to offer explainability. The dataset includes textual claims, chatgpt paraphrased claims, associated images, generated visual paraphrases, heatmap for explainability, and adversarial fake news stories.

**Reasons To Accept:**

The dataset is thorough and provides inspiration to the community for the following up study.

The paper is well-written.

**Reasons To Reject:**

No particular reason to reject it.

It would be better to conduct human evaluation also on the heatmap for more accurate and human-centered interpretation.

**Reproducibility:**

3: Could reproduce the results with some difficulty. The settings of parameters are underspecified or subjectively determined; the training/evaluation data are not widely available.

**Reviewer Confidence:**

3: Pretty sure, but there's a chance I missed something. Although I have a good feel for this area in general, I did not carefully check the paper's details, e.g., the math, experimental design, or novelty.

---

> ### Author Rebuttal · Authors · 2023-08-28
>
> Thank you for your review, please find our detailed response below -
>
> Reasons to reject:
>
> ---------------------------
>
> “conduct human evaluation also on the heatmap” -
>
> As discussed in section 4.3, pg. 5/35 in the main paper and section E.2 pg. 21/35 in the Appendix, we have reported our evaluation of the heatmap generation with Fréchet inception distance (FID) and  Mean Opinion Score (MOS) metrics. MOS is a  numerical measure of the human-judged perceived quality of artificially generated media. 10 human raters were assigned the task of evaluating MOS scores for 500 samples, as reported in fig. 7 on pg. 6/35.

---

### Official Review · Reviewer_c4E1 · 2023-08-05

**Typos Grammar Style And Presentation Improvements:** The sentences at lines 337 and 350 ap…
**Soundness:** 3

**Excitement:**

4: Strong: This paper deepens the understanding of some phenomenon or lowers the barriers to an existing research direction.

**Paper Topic And Main Contributions:**

This paper focuses on multimodal fact verification. The authors have proposed FACTIFY 3M, a large-scale multimodal fact verification dataset. Specifically, they collected data from the web and existing datasets. Each sample in the dataset includes a claim, an associated image, and an evidence document. Additionally, they employed data augmentation techniques, such as using ChatGPT for textual paraphrasing, Stable Diffusion for visual paraphrasing, and generating heatmaps for image explanation. For the interpretability of fact-checking, they also generated 5W pairs (who, what, when, where, and why). The QA process helps determine which part of the claim is inconsistent with the facts. Furthermore, they evaluated the quality of generated text and images to ensure the dataset's effectiveness and trained baseline models for comparison.

**Questions For The Authors:**

1.	Adversarial attacks can also be applied to the support class. Why only target the false class?
2.	Has there been any manual evaluation for the use of similarity-based classification?


**Reasons To Accept:**

1. Distinguishing malicious multimodal fake news has indeed become an urgent task. The FACTIFY 3M dataset proposed in this paper provides a large-scale benchmark for this task and facilitates further research.

2. This paper utilizes cutting-edge technologies such as ChatGPT and Stable Diffusion for data augmentation, and generated heatmaps and 5W pairs to achieve interpretability, which is somewhat novel.

3. The paper is well written and structured.


**Reasons To Reject:**

1. The authors claim to be the first ones to introduce adversarial attacks. However, there have been related works (e.g., https://arxiv.org/abs/2209.03755, https://arxiv.org/abs/2202.09381) in the field of fact-checking that have also introduced adversarial attacks.

2. The authors classify the collected data into support or neutral categories based on similarity, but they did not evaluate the classification accuracy, which I think is not convincing enough.


**Reproducibility:**

3: Could reproduce the results with some difficulty. The settings of parameters are underspecified or subjectively determined; the training/evaluation data are not widely available.

**Reviewer Confidence:**

4: Quite sure. I tried to check the important points carefully. It's unlikely, though conceivable, that I missed something that should affect my ratings.

---

> ### Author Rebuttal · Authors · 2023-08-28
>
> Thank you for your review, please find our detailed response below -
>
>
> Reasons to reject:
>
> ---------------------------
>
> “claim to be the first ones to introduce adversarial attacks” -
>
> Thank you for bringing these two papers to our notice. We will cite them and acknowledge that those previous research have already discussed adversarial attacks on fact-verification systems. We would like to extend the necessary discussion in the paper utilizing the extra page granted upon acceptance.
>
> However, we maintain that this aspect does not diminish the significance of this paper, which emphasized not only adversarial attack, but introducing the overall FACTIFY 3M dataset consists of -  (i) textual claims, (ii) ChatGPT-generated paraphrased claims, (iii) associated images, (iv) stable diffusion generated additional images (i.e., visual paraphrases), (v) pixel-level image heatmap to foster image-text explainability of the claim, (vi) 5W QA pairs, and (vii) adversarial fake news stories.
>
> A nuanced point to be noticed here is the referenced paper [2], which was made public only a week prior to the EMNLP submission deadline. Naturally we missed it, but will add necessary discussion in the final version given one extra page grant.
>
> References
>
> -----------------
>
> [1] Du, Y., Bosselut, A., & Manning, C. D. (2022, June). Synthetic disinformation attacks on automated fact verification systems. In Proceedings of the AAAI Conference on Artificial Intelligence (Vol. 36, No. 10, pp. 10581-10589). https://arxiv.org/pdf/2202.09381.pdf
>
> [2] Abdelnabi, S., & Fritz, M. (2023). {Fact-Saboteurs}: A Taxonomy of Evidence Manipulation Attacks against {Fact-Verification} Systems. In 32nd USENIX Security Symposium (USENIX Security 23) (pp. 6719-6736). https://arxiv.org/pdf/2209.03755.pdf
>
> “did not evaluate the classification accuracy”
>
> We have conducted an evaluation on a randomly sampled dataset of 1000 samples which was ongoing during submission but has recently concluded. Evaluation results affirm that our claims still sustain the accuracy. We will add this discussion in the final version.
>
>
> Questions For The Authors:
>
> -----------------------------------------
>
> “Adversarial attacks can also be applied to the support class. Why only target the false class?” -
> This is a very good point. We have also discussed within our team to keep this exploration for future tasks given the wide gamut of areas this paper touches upon. In this work, “we attempt to confuse the fact verification system by injecting fake examples acting as an adversarial attack.”
>
> “Has there been any manual evaluation for the use of similarity-based classification?” -
> We have conducted an evaluation on a randomly sampled dataset of 1000 samples which was ongoing during submission but has recently concluded. Evaluation results affirm that our claims still sustain the accuracy. We will add this discussion in the final version.

---

### Official Review · Reviewer_Nv99 · 2023-08-07

**Soundness:** 4

**Excitement:**

4: Strong: This paper deepens the understanding of some phenomenon or lowers the barriers to an existing research direction.

**Paper Topic And Main Contributions:**

The paper introduces a novel and valuable dataset named FACTIFY 3M for multimodal fact verification. The paper addresses the critical issue of disinformation and its impact on society. It highlights the need for efficient fact verification, especially in the context of multimodal disinformation, given the abundance of images and videos shared on social media platforms. The FACTIFY 3M dataset comprises 3 million samples and includes textual claims, paraphrased claims generated by ChatGPT, associated images, pixel-level image heatmaps, 5W question-answering pairs, and adversarial fake news stories. The dataset aims to facilitate research on various aspects of multimodal fact verification and promote explainability through visual question-answering. The paper emphasizes the dataset's potential to help detect adversarial attacks, verify complex facts, and aid journalists in fact-checking.

**Reasons To Accept:**

The paper's strength lies in the introduction of FACTIFY 3M, a comprehensive and large-scale dataset for multimodal fact verification. The dataset's diverse features, including textual claims, paraphrased claims, images, visual paraphrases, and 5W question-answering pairs, provide a rich resource for researchers to explore various aspects of fact verification. The incorporation of pixel-level image heatmaps for image-text explainability enhances the interpretability of fact verification decisions. By addressing the challenge of multimodal disinformation, the paper makes a significant contribution to combating the societal crisis of disinformation and its consequences.

**Reasons To Reject:**

While the paper highlights the importance of the proposed dataset, it lacks in-depth experimental evaluations or case studies using FACTIFY 3M. The authors should consider providing concrete examples and use cases to demonstrate the dataset's potential applications and showcase its utility for multimodal fact verification tasks.

**Reproducibility:**

3: Could reproduce the results with some difficulty. The settings of parameters are underspecified or subjectively determined; the training/evaluation data are not widely available.

**Reviewer Confidence:**

3: Pretty sure, but there's a chance I missed something. Although I have a good feel for this area in general, I did not carefully check the paper's details, e.g., the math, experimental design, or novelty.

---

> ### Author Rebuttal · Authors · 2023-08-28
>
> Thank you for your review, please find our detailed response below -
>
> Reasons to reject:
>
> -------------------------
>
> “lacks in-depth experimental evaluations or case studies using FACTIFY 3M”
>
> Throughout the paper, we have attempted to justify with concrete, illustrative examples why each aspect of the dataset is important and its potential applications and use cases. Specifically, in section 7 (page nos. 7-8), we have presented baseline models for: multimodal entailment and 5W QA-based validation. Table 2 on pg. 8/35 shows results of our model’s performance before and after adversarial injections using our curated FACTIFY3M dataset. Please clarify if you have any specific suggestion for case studies.

---

### Meta-Review · Senior_Area_Chairs · 2023-10-04

**Recommendation:** 4

**Metareview:**

The paper "FACTIFY 3M: A Multimodal Dataset for Fact Verification" introduces an intriguing and valuable contribution to the field of fact verification. It addresses the pressing issue of disinformation and its pervasive impact on society, particularly in the context of multimodal disinformation disseminated through images and videos on social media platforms. The dataset, FACTIFY 3M, is the central focus of the paper, consisting of 3 million samples encompassing textual claims, paraphrased claims, associated images, pixel-level image heatmaps, 5W question-answering pairs, and adversarial fake news stories. The primary objective of FACTIFY 3M is to support research endeavors related to multimodal fact verification and enhance interpretability through visual question-answering. The paper also emphasizes the dataset's potential applications, including detecting adversarial attacks, verifying complex facts, and assisting journalists in fact-checking.

In summary, the paper introduces a promising and relevant dataset in the field of multimodal fact verification. Its strengths lie in its comprehensiveness, the critical societal issue it addresses, and its focus on interpretability. However, to strengthen its case for acceptance, the paper should consider augmenting its content with concrete examples and experimental results demonstrating the practical value of FACTIFY 3M for researchers and practitioners engaged in fact verification tasks.

---

### Decision · Program_Chairs · 2023-10-07

**Decision:**

Accept-Main

**Comment:**

The paper "FACTIFY 3M: A Multimodal Dataset for Fact Verification" introduces an intriguing and valuable contribution to the field of fact verification. It addresses the pressing issue of disinformation and its pervasive impact on society, particularly in the context of multimodal disinformation disseminated through images and videos on social media platforms. The dataset, FACTIFY 3M, is the central focus of the paper, consisting of 3 million samples encompassing textual claims, paraphrased claims, associated images, pixel-level image heatmaps, 5W question-answering pairs, and adversarial fake news stories. The primary objective of FACTIFY 3M is to support research endeavors related to multimodal fact verification and enhance interpretability through visual question-answering. The paper also emphasizes the dataset's potential applications, including detecting adversarial attacks, verifying complex facts, and assisting journalists in fact-checking.

In summary, the paper introduces a promising and relevant dataset in the field of multimodal fact verification. Its strengths lie in its comprehensiveness, the critical societal issue it addresses, and its focus on interpretability. However, to strengthen its case for acceptance, the paper should consider augmenting its content with concrete examples and experimental results demonstrating the practical value of FACTIFY 3M for researchers and practitioners engaged in fact verification tasks.